# Multi-faceted immunomodulatory and tissue-tropic clinical bacterial isolate potentiates prostate cancer immunotherapy

Jonathan F. Anker [ID] [1], Anum F. Naseem[1], Hanlin Mok[1], Anthony J. Schaeffer[1], Sarki A. Abdulkadir[1,2,3] & Praveen Thumbikat[1,3]

Immune checkpoint inhibitors have not been effective for immunologically "cold" tumors, such as prostate cancer, which contain scarce tumor infiltrating lymphocytes. We hypothesized that select tissue-specific and immunostimulatory bacteria can potentiate these immunotherapies. Here we show that a patient-derived prostate-specific microbe, CP1, in combination with anti-PD-1 immunotherapy, increases survival and decreases tumor burden in orthotopic *MYC*- and *PTEN*-mutant prostate cancer models. CP1 administered intraurethrally specifically homes to and colonizes tumors without causing any systemic toxicities. CP1 increases immunogenic cell death of cancer cells, T cell cytotoxicity, and tumor infiltration by activated CD8 T cells, Th17 T cells, mature dendritic cells, M1 macrophages, and NK cells. CP1 also decreases intra-tumoral regulatory T cells and VEGF. Mechanistically, blocking CP1-recruited T cells from infiltrating the tumor inhibits its therapeutic efficacy. CP1 is an immunotherapeutic tool demonstrating how a tissue-specific microbe can increase tumor immunogenicity and sensitize an otherwise resistant cancer type to immunotherapy.

[1] Department of Urology, Northwestern University Feinberg School of Medicine, Chicago, IL 60611, USA. [2] The Robert H. Lurie Comprehensive Cancer Center, Northwestern University Feinberg School of Medicine, Chicago, IL 60611, USA. [3] Department of Pathology, Northwestern University Feinberg School of Medicine, Chicago, IL 60611, USA. These authors contributed equally: Sarki A. Abdulkadir, Praveen Thumbikat. Correspondence and requests for materials should be addressed to S.A.A. (email: Sarki.abdulkadir@northwestern.edu) or to P.T. (email: thumbikat@northwestern.edu)

mmune checkpoint inhibitors have shown great promise in recent years, with anti-PD-1/PD-L1 and anti-CTLA-4 blocking antibodies gaining approval in multiple cancer types. The efficacy of these immunotherapies in overcoming tumor-driven immunosuppression is largely dependent on the level of tumor infiltrating lymphocytes (TILs) at the time of treatment[1]. Interestingly, tumor mutagenicity has been linked to tumor immunogenicity, due to the generation of tumor-specific neo-antigens, and, as a result, has been associated with immune checkpoint inhibitor efficacy[2]. However, many cancers do not contain the high mutational rate seemingly required to receive clinical benefit[3]. This includes prostate cancer, the most prevalent non-skin cancer in men responsible for the second most cancer deaths[4]. Consequently, to date, immune checkpoint inhibitors have largely failed to produce clinical benefit for the disease, with ipilimumab (anti-CTLA-4)[5] and nivolumab (anti-PD-1)[6] monotherapies providing no improvement in overall survival for patients with castration-resistant prostate cancer (CRPC), despite the majority of prostate tumors express high PD-L1 levels[7] and the few TILs[8] being highly PD-1 positive[9,10].

To optimize immunotherapy efficacy, focus has shifted toward combining immune checkpoint inhibitors with various external therapeutics, such as adoptive T cell therapies, chemotherapies, and radiation[11]. However, these combinations often are not specific or optimal for the tissue-type being treated. Rather than utilizing exogenous agents, endogenous agents found within each of these tissues represent a more personalized and optimized therapeutic option. Microbes, specifically pathogenic bacteria, colonize various tissue niches throughout the human body[12–15]. We hypothesize that select clinically derived tissue-specific bacteria that exhibit innate tissue-tropism and local immunostimulatory properties can be utilized to enhance immunotherapies in resistant tumor types.

One such microbe is CP1, a uropathogenic *Escherichia coli* (UPEC) isolated from the prostatic secretions of a patient with chronic prostatitis without concurrent cystitis. Intra-urethrally administered CP1 specifically colonizes murine prostates before being cleared by the host after 1 month. During that time, CP1 induces local, tissue-specific Th1/Th17 T cell infiltration[16,17]. In addition, after administration to pre-cancerous genetically susceptible mice, CP1 induced chronic inflammation, and therefore, expectedly, modestly increased the frequency of progression from mouse prostatic intraepithelial neoplasia (mPIN)[18]. As increasing research has demonstrated the context-dependent balance between chronic inflammation promoting cancer formation and the anti-tumor immune response combating tumor growth, we hypothesized that, in the setting of a developed immunosuppressive tumor, the immunostimulatory CP1 could be exploited on the other side of the spectrum to drive a therapeutic anti-prostate tumor immune response.

Here we demonstrate in multiple clinically relevant orthotopic models of prostate cancer that CP1 homes to and colonizes tumors, induces infiltration by multiple anti-tumor immune cell types, increases tumor immunogenicity, and decreases immunosuppressive immune cell types and molecules within the tumor microenvironment, resulting in strong clinical benefit in combination with PD-1 blockade. In addition to treating prostate cancer, these results outline the potential to discover additional unique tissue-specific bacteria to benefit patients with other immunologically "cold" cancers.

## Results

### CP1 is a patient-derived UPEC that homes to prostate tumors.
CP1 is a clinically derived *E. coli* from a patient with chronic prostatitis that has previously been shown able to colonize murine

prostates and induce a tissue-specific local inflammatory response[17]. To further characterize the bacteria, we performed whole-genome sequencing, which revealed that CP1 contains a 5,841,456 base pair genome with 50.9% GC content and 5172 unique coding sequences, 74 unique rRNA sequences, and 95 unique tRNA sequences (Supplementary Figure 1a). Further, CP1 is categorized within the B2 phylogenetic group (Supplementary Figure 1b) and sequencing type 131 (ST131). Phylogenetic tree analysis grouped CP1 closely with other UPEC isolates, including CFT073, UTI89, 536, J96, and NA114. Interestingly, CP1 is an atypical ST131 *E. coli*, as it lacks multiple consensus virulence genes (each with ≥93% ST131 population prevalence). While CP1 contained identical multi-locus sequence typing (MLST) alleles as the ST131 NA114 strain, only NA114 contained all consensus virulence factor genes (Supplementary Figure 1b)[19], indicating that CP1 is potentially less virulent than other similar UPECs.

UPECs are able to colonize the urinary tract and invade and proliferate within host epithelial cells[20], and prior analysis of CP1 demonstrated that it is able to adhere to and invade benign prostate epithelial cell lines[17]. To test if CP1 could invade prostate cancer cells, we repeated this in vitro gentamicin protection assay with the *MYC*-driven murine prostate cancer cell line, Myc-CaP[21]. As a control, we utilized MG1655, the prototypical strain of the patient-derived K-12 *E. coli* isolate that has been maintained with "minimal genetic manipulation" and whose complete genome has been sequenced[22]. About 19.7% of the genes in CP1 were not present in the MG1655 genome, and the remaining shared genes contained an average 93.9% identity (Supplementary Figure 1c). As with the benign prostate epithelial cell lines, CP1 was able to adhere to, invade, and intracellularly proliferate within Myc-CaP cells, and did so to a greater degree than did MG1655 (Supplementary Figure 2).

A prior study has established that intra-urethral instillation of $2 \times 10^8$ CP1 in mice leads to bacterial colonization of the benign prostate, and, to a lesser degree, the bladder, thereby recapitulating the common natural ascending pattern of prostatic infection in humans[17]. To similarly evaluate CP1 in a clinically relevant in vivo model of prostate cancer, we injected Myc-CaP cells intra-prostatically, leading to orthotopic prostate tumor development. Eight days after intra-prostatic injection, mice with established tumors were intra-urethrally administered $2 \times 10^8$ CP1. Tissue analysis 9 days after CP1 administration revealed that CP1 specifically colonized prostate tumor tissue, ascending from the urethra to the bladder to the tumor without progressing to the kidneys or colonizing systemic tissues (Fig. 1a). An average $3.8 \times 10^6$ total CP1 colony forming units (CFUs) (Fig. 1a), or $3.3 \times 10^6$ CFU/g tumor (Fig. 1b), were cultured from tumors, representing approximately 1.9% of the initial CP1 inoculation (Fig. 1c). Additional comparison of CP1 tumor colonization on day 1 and day 9 after intra-urethral administration revealed no significant changes in CFUs over time (Supplementary Figure 3a-c). We also analyzed bacterial *16S* RNA from tumor tissue as an additional means of tracking intra-tumoral CP1. As expected, *16S* RNA levels were higher in CP1-administered tumors (Fig. 1d). Calibrating *16S* RNA values to CP1 cell counts resulted in similar values as those attained by tumor tissue culture at both timepoints (Supplementary Figure 3d), suggesting that viable but non-culturable (VBNC) CP1 were absent or minimal in this model. Finally, immunofluorescent analysis of tumor tissue 9 days after intra-urethral CP1 administration identified the presence of both extracellular (approximately 58.2%) and intracellular (approximately 41.8%) *E. coli* throughout the tumors (Fig. 1e). Importantly, CP1 administration did not cause any systemic toxicities, with no changes in body weight or any serum chemistry laboratory values, and all complete blood count (CBC) values fell within their normal range (Supplementary Figure 4), other than

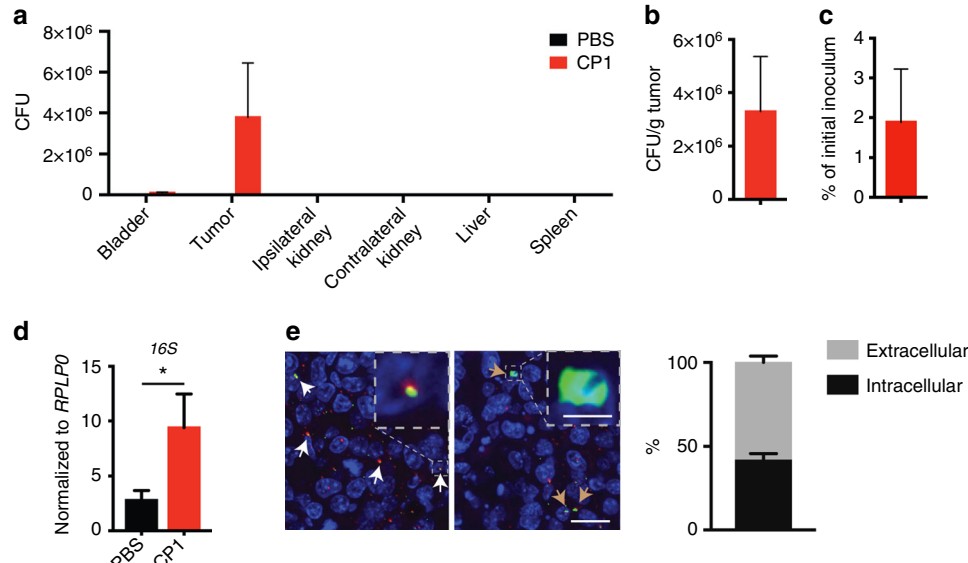

**Fig. 1** CP1 specifically homes to and colonizes prostate tumor tissue. Orthotopic prostate tumor-bearing mice were analyzed 9 days after intra-urethral CP1 administration. **a** Bacterial colonization in the bladder, prostate tumor (also represented as **b** CFU/g and **c** percentage of the initial $2 \times 10^8$ CP1 intra-urethral inoculum), ipsilateral and contralateral kidneys, liver, and spleen, performed in biological triplicates, plated in serial dilutions. **d** 16S qRT-PCR of tumor RNA, normalized to RPLP0, performed in biological quadruplicates, technical duplicates. **e** E. coli IF of tumor tissue (yellow/red = extracellular, indicated with white arrows; green = intracellular, indicated with brown arrows) (scale bar, 20 μm; magnified scale bar, 4 μm). Mice $n = 4$–5/group, performed in two independent experiments, E. coli IF quantified with quadruplicate FOVs/tumor. Data represented as mean ± S.E.M. Statistical significance was determined by two-tailed Student's t-test. $*P < 0.05$

low red blood cell distribution width (RDW), which is clinically insignificant in the absence of anemia. Thus, intra-urethrally administered CP1 specifically and safely colonized prostate tumor tissue.

**CP1 induces immunogenic cell death (ICD) and pro-inflammatory cytokine production.** Interestingly, in vitro culture of Myc-CaP cells with CP1 resulted in cancer cell death in a dose-dependent manner (Supplementary Figure 5a). Therefore, we analyzed whether this was specifically ICD. All three major ICD damage-associated molecular patterns (DAMPs), HMGB1, ATP, and calreticulin[23], were elevated in the presence of live, but not heat killed, CP1 (Fig. 2a). Similar results were seen with human LNCaP prostate cancer cells (Fig. 2b). CP1 also induced all ICD markers to a significantly higher level than did MG1655 (Supplementary Figure 5b, CP1 vs. MG1655: HMGB1 $P = 0.0011$, ATP $P = 0.009$, calreticulin $P = 0.0014$ all by two-tailed Student's t-test). To more accurately represent the quantity of CP1 present within the tumor, the in vitro ICD assays were repeated with the addition of gentamicin at a multiplicity of infection (MOI) of 1. These conditions resulted in a final average CP1:Myc-CaP ratio of 0.005, with the surviving intracellular CP1 representing approximately 10.9% of the initial bacteria added to the culture (multiple orders of magnitude less bacteria than without gentamicin). In the presence of gentamicin, CP1 still significantly increased the percent of calreticulin+ Myc-CaP cells, but did not induce HMGB1 or ATP secretion (Supplementary Figure 5c). However, it is important to note that in addition to decreasing total CP1 count, gentamicin also eliminated any potential importance of extracellular CP1 interacting with tumor cells or CP1 spreading between cells. Finally, we tested for ICD within tumor tissue 9 days after intra-urethral CP1 administration. CP1-administered prostate tumors contained an increased percentage of HMGB1− nuclei (Fig. 2c), signifying HMGB1 release[24], and areas of increased cell surface calreticulin levels (Fig. 2d).

Further analysis of cell death pathways identified that CP1 significantly increased caspase 3/7 activity without gentamicin (Supplementary Figure 5d, $P = 0.0103$ by two-tailed Student's t-test, CP1 significantly greater than MG1655, $P = 0.0037$ by two-tailed Student's t-test) and with gentamicin (Supplementary Figure 5e ($P = 0.0286$ by two-tailed Student's t-test), f). CP1 exposed Myc-CaP cells also displayed an increased late apoptotic phenotype (Annexin V+ PI+) without gentamicin (Supplementary Figure 5g) and an increased early apoptotic phenotype (Annexin V+ PI−) with gentamicin (Supplementary Figure 5h, CP1 significantly greater than MG1655, $P = 0.0329$ by two-tailed Student's t-test). However, no changes were observed in the phosphorylation of MLKL, RIP1 levels, or PARP cleavage after Myc-CaP culture with either CP1 or MG1655 and gentamicin (Supplementary Figure 5i), suggesting that CP1-induced ICD is occurring in a necroptosis-independent manner[24].

Finally, CP1 significantly increased the in vitro Myc-CaP production of pro-inflammatory cytokines and chemokines IL-9, IL-15, IL-1α, IFNγ, MIP-2, MIP-1β, G-CSF, IL-17, KC, IL-2, and IP-10 (CXCL10), which is important for ICD, while also decreasing vascular endothelial growth factor (VEGF) from cancer cells (Fig. 2e). Overall, CP1 induced ICD, increased pro-inflammatory cytokines and chemokines, and decreased VEGF from cancer cells.

**CP1 increases TILs and reprograms the tumor microenvironment.** To evaluate CP1's ability to remodel the "cold" prostate tumor microenvironment, we immunophenotyped tumors 9 days after intra-urethral bacterial administration. CP1 increased T cells not only in the tumor stroma and periphery, but also intra-tumorally (Fig. 3a), consisting of both CD8 and CD4 TILs (Fig. 3b). In contrast, intra-urethral MG1655 administration did not result in increased TILs (Supplementary Figure 6). Further analysis revealed that the increased CD8 TILs in CP1-administered tumors expressed increased TNFα (Fig. 3c) and the activation marker PD-1 (Fig. 3d), and a higher percentage

expressed IFNγ within the tumor draining lymph nodes (dLNs) (Fig. 3e). In addition, intra-tumoral (Fig. 3f) and dLN (Fig. 3g) CD4 T cells were Th17-polarized. CP1 administration also

decreased the percentage of regulatory T cell (Treg) TILs, with most tumors containing a >3-fold increased CD8/Treg ratio (Fig. 3h). Notably, despite increasing overall hematopoietic

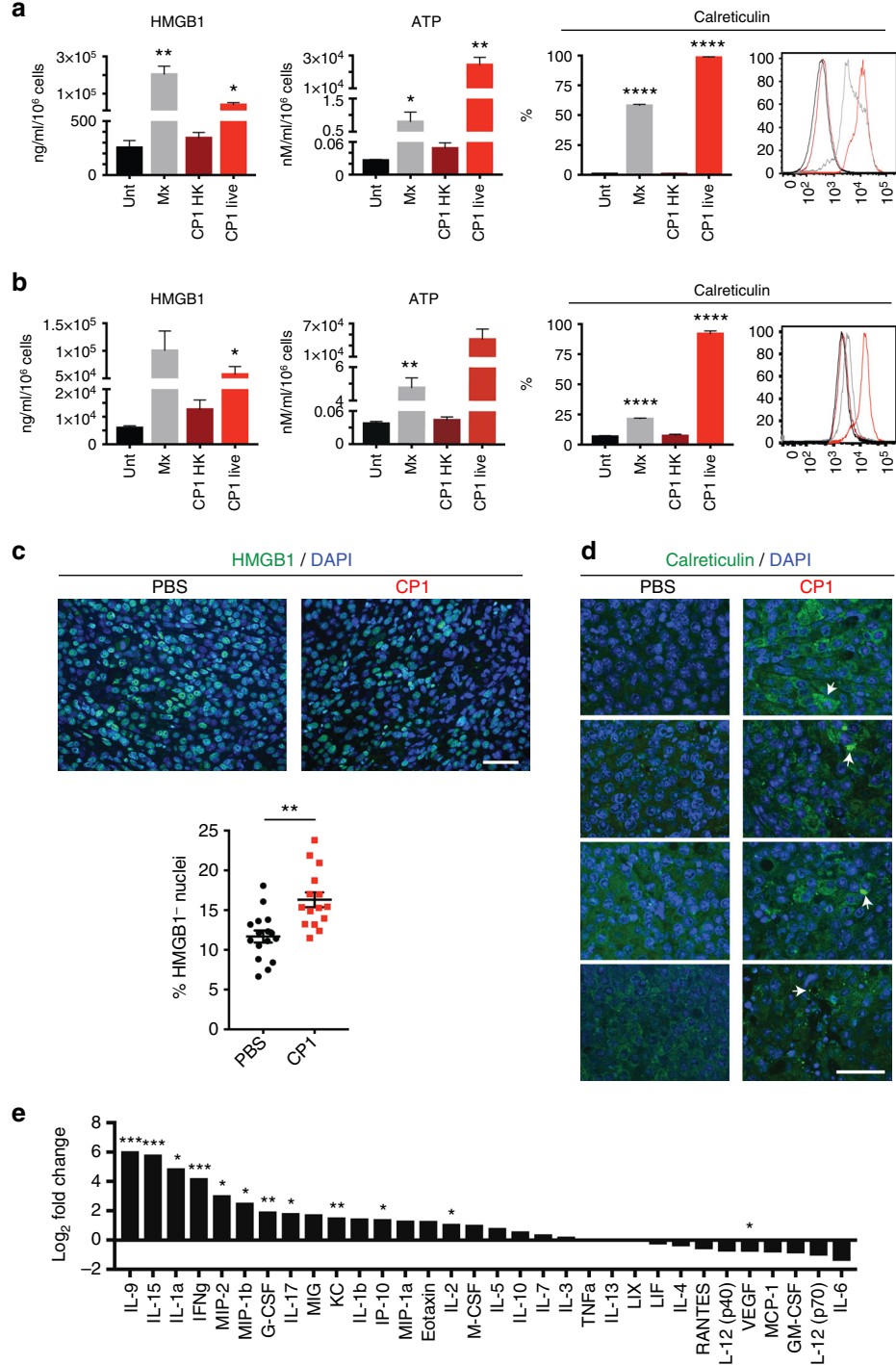

**Fig. 2** CP1 induces immunogenic cell death while increasing pro-inflammatory cytokines and chemokines and decreasing VEGF. ICD was assessed in vitro from co-culture of **a** Myc-CaP or **b** LNCaP cells with untreated (Unt.), mitoxantrone (Mx), heat killed (HK) CP1, or live CP1 via HMGB1 (ELISA), ATP (luminescence assay), and calreticulin (flow cytometry, with representative histogram (Unt = black, Mx = gray, CP1 HK = dark red, CP1 live = red)), performed in biological triplicates, technical duplicates, statistics compared to Unt. ICD was assessed in vivo by **c** HMGB1 or **d** calreticulin IF of prostate tumor tissue 9 days after intra-urethral CP1 administration, with representative images (each calreticulin image representative of a different tumor with white arrows indicating foci of cell surface staining, green = HMGB1 or calreticulin, scale bar, 50 μm). Mice n = 4/group, HMGB1 quantified with quadruplicate FOVs/tumor. **e** Multiplex cytokine and chemokine array from Myc-CaP supernatant, performed in biological triplicates, technical duplicates. Data represented as mean ± S.E.M. or log₂ fold change with and without CP1 exposure. Statistical significance was determined by two-tailed Student's t-test (**a**, **b**, each group compared to Unt.). *P < 0.05, **P < 0.01, ***P < 0.001, ****P < 0.0001

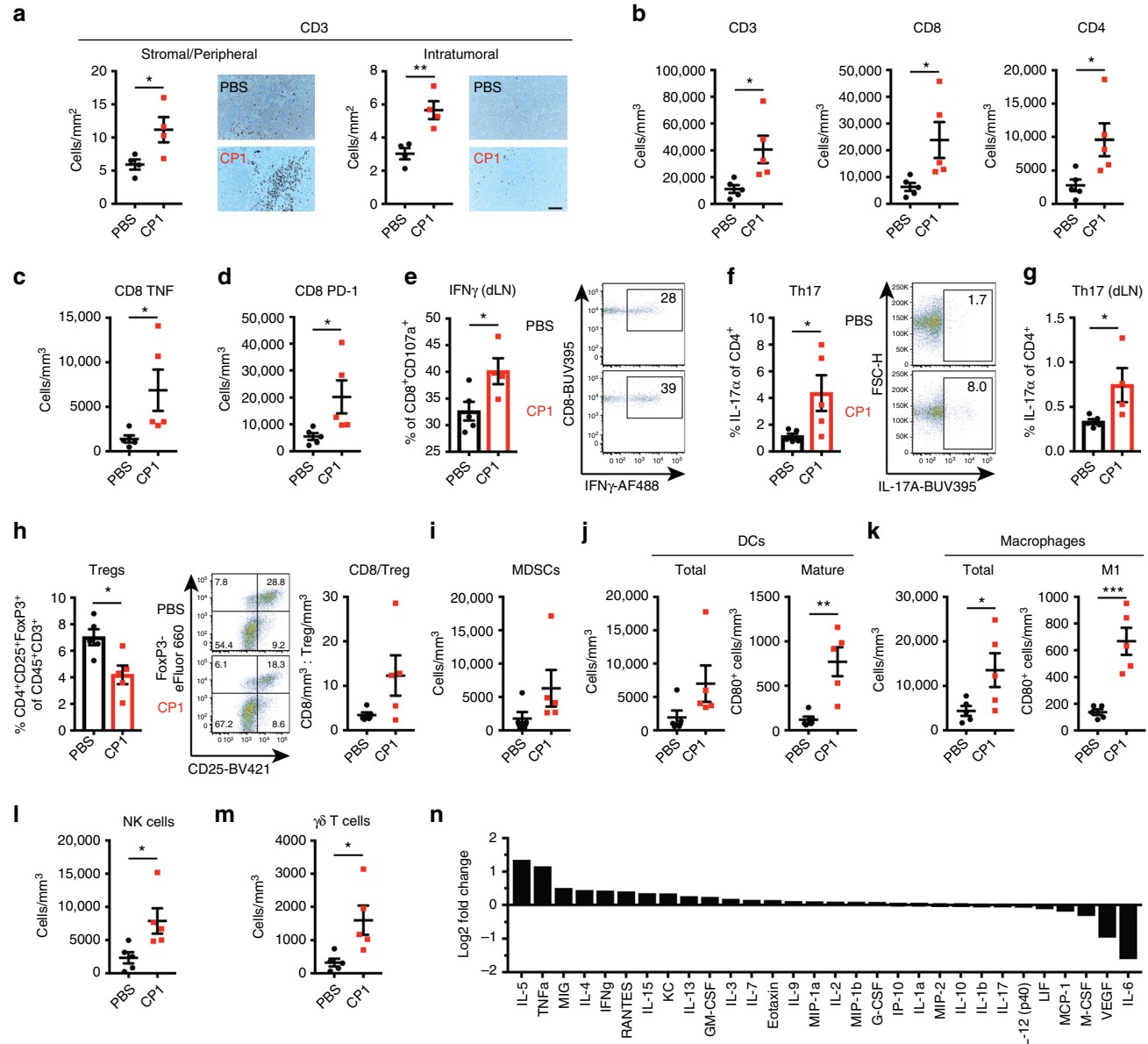

**Fig. 3** CP1 increases TILs and tumor immune infiltration while decreasing Tregs. **a** Blinded IHC with representative images (scale bar, 100 µm) and **b–m** flow cytometry analysis of Myc-CaP tumors or dLNs, as indicated, displayed as cell counts normalized to tumor volume (scatter plots) or percentages of parent gate (scatter boxed plots), with representative flow cytometry plots. MDSCs were defined as CD11b$^+$Gr-1$^+$. **n** Multiplex cytokine and chemokine array from Myc-CaP tumors. Mice $n = 4$–5/group, performed in two independent experiments. Data represented as mean ± S.E.M. or log$_2$ fold change with and without CP1 administration. Statistical significance was determined by two-tailed Student's $t$-test. *$P < 0.05$, **$P < 0.01$, ***$P < 0.001$

infiltration, CP1 did not increase the infiltration of myeloid-derived suppressor cells (MDSCs; CD11b$^+$Gr-1$^+$) (Fig. 3i). Interestingly, CP1 significantly increased both mature dendritic cells (DCs) and M1-polarized macrophages to a much greater degree than either total cell type (Fig. 3j,k), while also increasing infiltration of NK cells (Fig. 3l), γδ T cells (Fig. 3m), and B cells (Supplementary Figure 7a). While CP1 did not increase PD-L1 on tumor or hematopoietic cells, the immune compartment was a greater source of PD-L1 within these tumors due to increased overall CD45$^+$ infiltration (Supplementary Figure 7b–d). IL-5 and TNFα were the most upregulated cytokines in CP1-treated tumors, and, consistent with the in vitro cytokine/chemokine array, IFNγ was among the most upregulated and IL-6 and VEGF among the most downregulated cytokines after CP1 administration (Fig. 3n). Overall, intra-tumoral CP1 increased infiltration of multiple anti-tumor immune cell types while decreasing Tregs.

**CP1 with PD-1 blockade is efficacious for prostate cancer.** To determine the functional implications of this immunomodulation, we treated orthotopic Myc-CaP tumor-bearing mice with intra-peritoneal anti-PD-1 antibody beginning 9 days after intra-urethral CP1 administration. We utilized pre-treatment in vivo bioluminescent imaging to normalize tumor burden and variance between experimental groups in this and all subsequent experiments (Supplementary Figure 8). To analyze survival, mice were followed after treatment termination with no additional interventions. Combination immunotherapy of CP1 and PD-1 blockade (CP1 + PD-1) significantly increased survival ($P = 0.0066$ by Log-rank test), conferring >2-fold increased 50% survival time. In contrast, neither CP1 nor anti-PD-1 monotherapy significantly enhanced survival (Fig. 4a). In mice analyzed immediately after treatment termination, CP1 + PD-1 decreased tumor burden, as assessed by in vivo bioluminescent imaging,

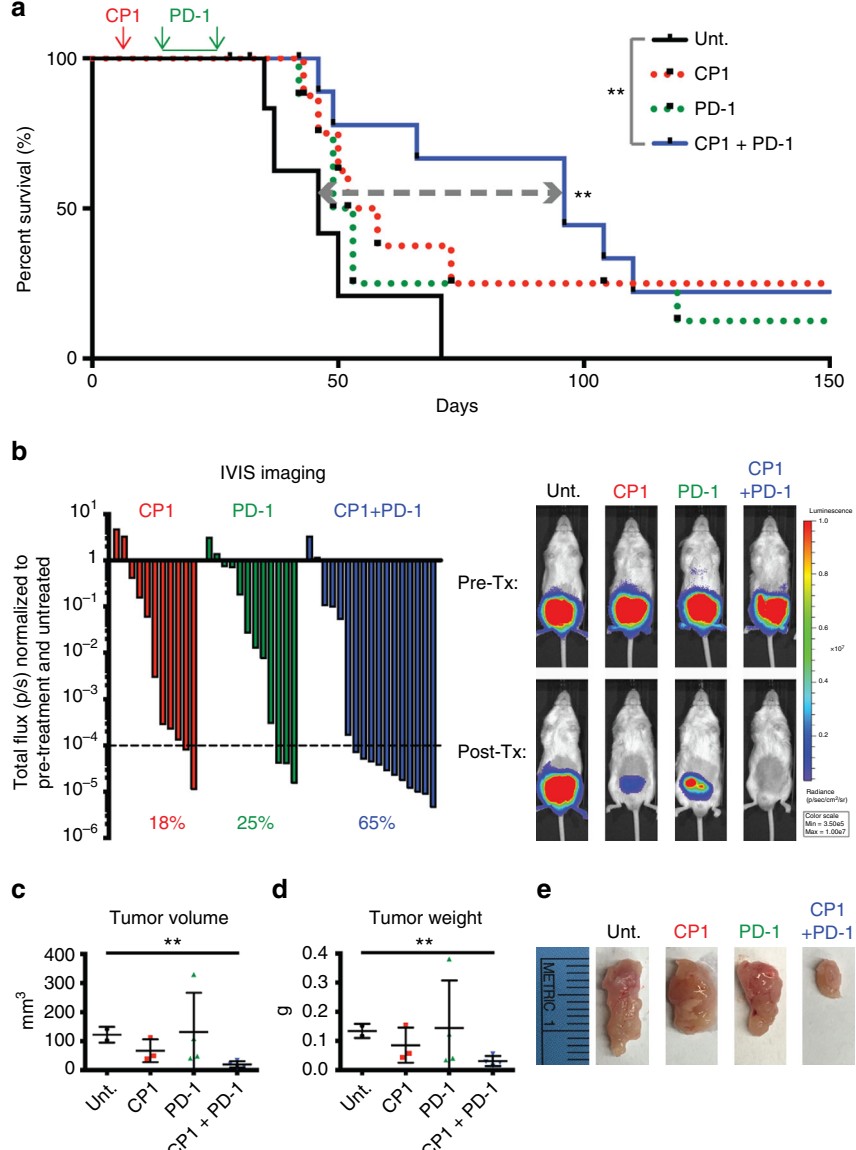

**Fig. 4** Combination CP1 and anti-PD-1 immunotherapy is efficacious in treating orthotopic prostate tumors. **a** Kaplan–Meier survival curve of Unt., CP1, anti-PD-1 antibody, or combination CP1 and anti-PD-1 antibody treated mice, mice $n = 6$–12/group, performed in three independent experiments. **b** Waterfall plot of IVIS imaging quantification, with each bar representing the post-treatment (Tx) total flux of a single tumor normalized to both its own pre-tx total flux and Unt. normalized total flux, with representative images. Percentages indicate the fraction of tumors with values <0.0001; $n = 11$–17 mice/experimental group, performed in four independent experiments. Post-tx tumor **c** volumes, **d** weights, and **e** gross images, mice $n = 3$–4/group. Data represented as mean ± S.E.M. Statistical significance was determined by **a** Log-rank test, **c**, **d** two-tailed Student's $t$-test. **$P < 0.01$

tumor weight, and tumor volume (Fig. 4b–e). In addition, CP1-treated tumors showed evidence of bacterial colonization, as determined by *16S* RNA (Supplementary Figure 9a). Interestingly, all long-term surviving mice (>75 days) contained relatively high *16S* RNA ratios (normalized to a mouse housekeeping gene) (Supplementary Figure 9b), suggesting that high bacterial load was important for therapeutic efficacy. In addition, as observed previously (Supplementary Figure 3), relative bacterial load did not increase over time (Supplementary Figure 9b). Further, within CP1-treated tumors, MIP-2 was the most upregulated and VEGF was again the most downregulated cytokine (Supplementary Figure 10a).

**CP1 with PD-1 blockade treats *PTEN*-deficient prostate cancer.** To challenge our combination immunotherapy in a second, more

aggressive and immunosuppressive model of advanced prostate cancer, we utilized CRISPR-Cas9 to knock out (KO) *PTEN* from the Myc-CaP genome. Loss of PTEN is linked to increased PD-L1 in prostate and breast cancer[25], and decreased TILs and resistance to PD-1 blockade in melanoma[26]. Further, concurrent copy number gain of *MYC* and loss of *PTEN* is associated with prostate cancer-specific mortality, reported in 57% of metastatic tumors compared to 9.6% in localized disease[27]. Similarly, this combination copy number alteration was present in 24.8% and 11.2% of SU2C/PCF metastatic and TCGA primary prostate tumors, respectively (Supplementary Figure 11). Myc-CaP PTEN KO cells contained increased phosphorylated-AKT and androgen receptor (Fig. 5a), and expressed approximately 2-fold higher levels of PD-L1, PD-L2, CD95, and CD95L, all important in tumor immune-evasion (Fig. 5b). These cells proliferated faster, particularly in

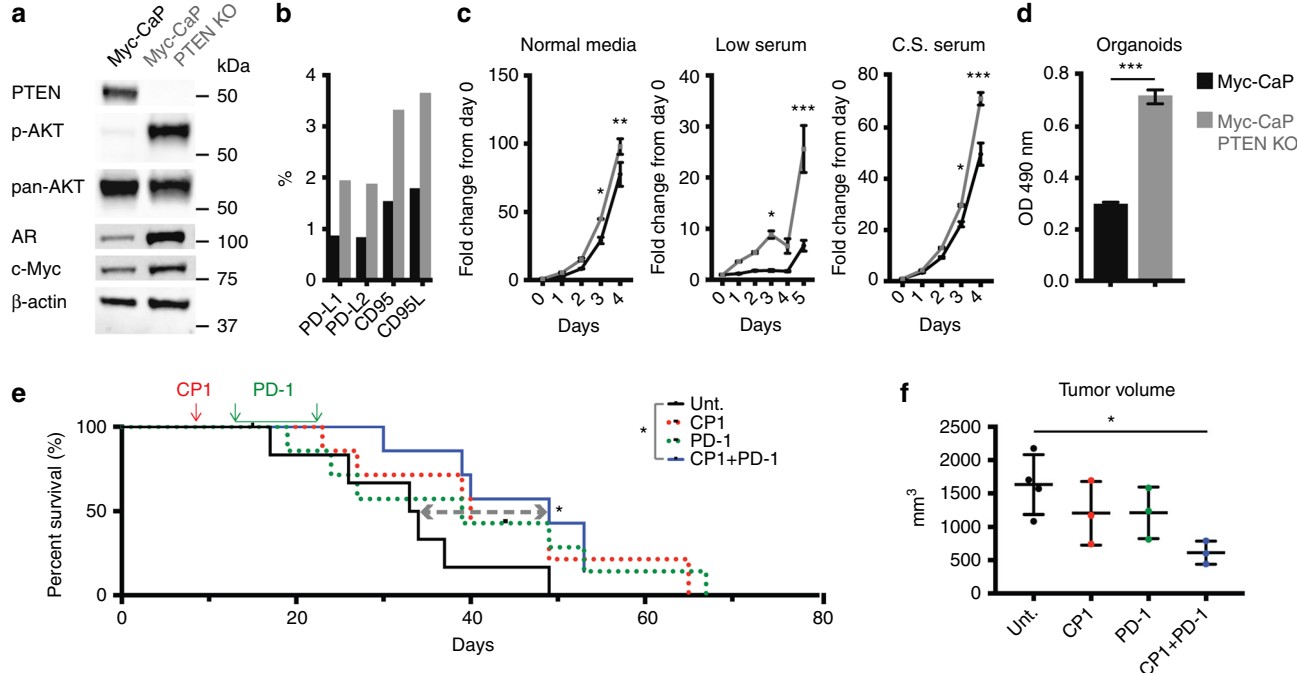

**Fig. 5** Combination CP1 and anti-PD-1 immunotherapy is efficacious in treating a novel orthotopic advanced prostate cancer model. In vitro comparison of Myc-CaP and Myc-CaP PTEN KO cell lines by **a** western blot (p-AKT = phosphorylated AKT, AR = androgen receptor), **b** flow cytometry, **c** growth rate by MTS assays in normal media (10%), low serum (1%), and charcoal stripped (C.S.) FBS, performed in triplicates, and as **d** 3-dimensional organoids, performed in sextuplicates. Myc-CaP PTEN KO **e** Kaplan–Meier survival curve, $n = 7$ mice/experimental group, performed in two independent experiments, and **f** tumor volumes, $n = 3–6$ mice/experimental group. Data represented as mean ± S.E.M. Statistical significance was determined by **c** two-way ANOVA, **d** two-tailed Student's $t$-test, **e** Log-rank test, **f** one-way ANOVA. *$P < 0.05$, **$P < 0.01$, ***$P < 0.001$

low and charcoal-stripped serum (Fig. 5c), and more rapidly formed 3-dimensional organoids (Fig. 5d). Thus, we generated a novel PTEN KO Myc-CaP cell line that displayed many characteristics of advanced human prostate cancer.

Orthotopic Myc-CaP PTEN KO tumor-bearing mice were again administered intra-urethral CP1 and subsequent anti-PD-1 antibody. As previously observed, neither CP1 nor anti-PD-1 monotherapy significantly increased survival of mice, whereas CP1 + PD-1 combination therapy conferred a significant 1.5-fold increased survival ($P = 0.0251$ by Log-rank test) (Fig. 5e). Upon analysis immediately after treatment termination, combination CP1 + PD-1 also significantly decreased tumor size (Fig. 5f, $P = 0.0344$ by one-way ANOVA). However, while tumor weight did not differ between groups (Supplementary Figure 12a), CP1 tumors were significantly denser (Supplementary Figure 12b), and contained increased exudate, as measured by fibrinogen (Supplementary Figure 12c). Therefore, tumor weight did not accurately assess therapeutic efficacy, consistent with pseudo-progressions observed with clinical immunotherapies[28].

**CP1 increases activated TILs and decreases Tregs and VEGF.** In Myc-CaP PTEN KO tumors, CP1 treatment again increased TILs (Fig. 6a), with degranulated T cells from CP1 and/or CP1 + PD-1 treated mice displaying increased cytotoxic functionality via IFNγ (Fig. 6b), granzyme B (Fig. 6c), and perforin (Fig. 6d) expression. Also consistent with its effects in Myc-CaP tumors, CP1 increased PD-1 on CD8 TILs (Fig. 6e), decreased the percentage of Treg TILs (Fig. 6f), and caused decreased VEGF and increased MIP-2, IL-17, and TNFα within the tumor microenvironment (Supplementary Figure 10b). Overall, combination CP1 and anti-PD-1 immunotherapy was efficacious in a second, more advanced model of the disease, increasing TILs and cytotoxic T cell function while decreasing Tregs and VEGF.

**CP1 therapeutic efficacy is dependent on recruitment of TILs.** To determine if CP1-recruited TILs were necessary for its immunotherapeutic efficacy, we utilized fingolimod (FTY720), a sphingosine-1 phosphate mimetic, to block egress of T cells from lymph nodes into peripheral tissues[29]. This approach did not inhibit the quantity or functionality of baseline TILs, thereby selectively blocking only those T cells recruited by CP1. FTY720 administration successfully blocked the CP1-dependent increase in both CD8 and CD4 TILs (Fig. 7a–d), and, consequently, reversed the anti-tumor efficacy of CP1 + PD-1 combination therapy (Fig. 7e, f). Therefore, TILs specifically recruited by CP1 were necessary to drive the anti-tumor immune response.

**Discussion**

Immune checkpoint inhibitors have thus far failed to provide significant clinical benefit for prostate cancer[5,6]. Similar to other immunologically "cold" and unresponsive tumor types, prostate tumors display strong PD-L1 positivity[7] and their microenvironment contains scarce[8] but high PD-1-expressing TILs[9,10], and high levels of Tregs[10,30] and M2-polarized tumor-associated macrophages (TAMs)[31], all of which are linked to disease progression and death. However, another similarity between these cancers is that they stem from tissues frequently colonized by pathogenic bacteria. This study demonstrates how a patient-derived prostate-specific bacteria was isolated and utilized to enhance immunotherapy efficacy in multiple orthotopic models of prostate cancer.

After intra-urethral administration, CP1 specifically homed to prostate tumors, ascending from the urethra to the bladder to the tumor without progressing to the kidneys or inducing any systemic toxicities. Further, CP1 induced ICD in both mouse and human prostate cancer cells in vitro, as determined by increased HMGB1, ATP, calreticulin, and CXCL10, and in tumor tissue

in vivo, as determined by HMGB1 secretion and increased cell surface calreticulin. ICD is important for the recruitment, activation, and optimization of antigen presentation of DCs[23]. Intra-tumoral CP1 also increased infiltration of CD8 and CD4 TILs, both of which are linked to response to PD-1 blockade[1], and T cells from these mice expressed increased levels of IFNγ, granzyme B, perforin, and TNFα, all critical molecules for a functional anti-tumor adaptive immune response[32]. In addition, CP1 decreased the Treg TIL phenotype with a corresponding increase in Th17 T cells. Tregs are a major source of immunosuppression in the tumor microenvironment[33], while IL-17 can promote anti-tumor immunity by increasing infiltration of CD8 TILs, NK cells, and APCs, as well as increasing IFNγ

production[34]. CP1 also increased intra-tumoral levels of mature DCs, M1-polarized macrophages, NK cells, and γδ T cells. Prostate TAMs are typically M2-polarized immunosuppressive cells linked to disease progression[31]. However, the CD80[+] TAMs recruited and polarized by CP1 represent an M1 macrophage proven to inhibit tumor growth and promote cytotoxic T cell activity[35]. Additionally, NK cells can directly kill cancer cells, secrete IFNγ, TNF, GM-CSF, and other cytokines to promote CD8 T cell and APC activity, while also controlling tumor metastasis and recognizing CD8 T cell-resistant tumors with downregulated MHC I[36]. γδ T cells can also directly eliminate cancer cells, produce IFNγ, and enhance CD8 T cell, Th1 T cell, and NK cell activity[37]. Finally, CP1 consistently increased the

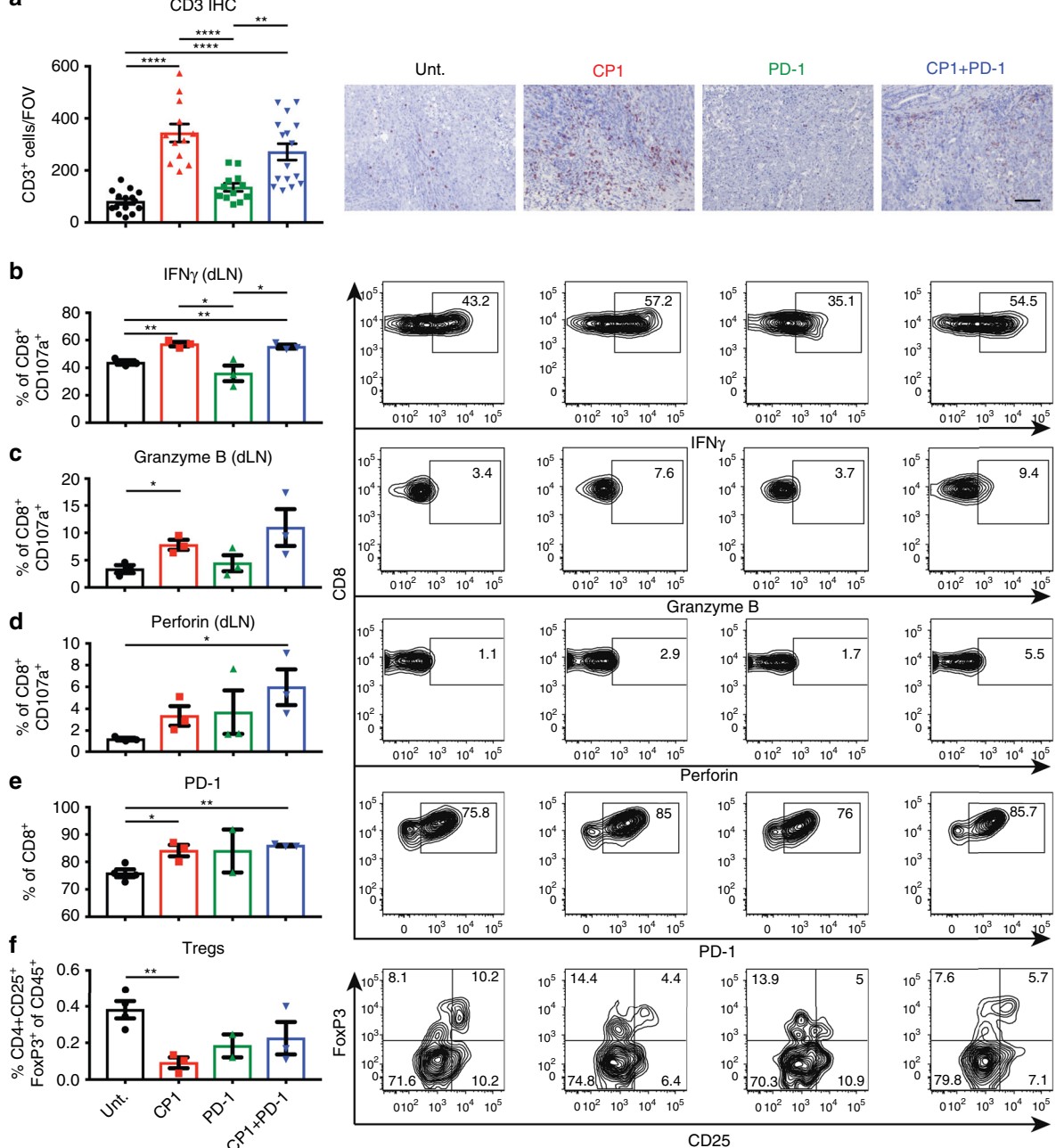

**Fig. 6** CP1 increases TILs and T cell cytotoxicity while decreasing Tregs in Myc-CaP PTEN KO tumors. **a** IHC with representative images, quadruplicate FOVs scored per sample (scale bar, 100 μm). **b–f** Flow cytometry analyses of tumors or dLNs with representative flow cytometry plots; $n = 3$–4 mice/experimental group. Data represented as mean ± S.E.M. Statistical significance was determined by **a** one-way ANOVA, **b–f** two-tailed Student's $t$-test. $*P < 0.05$, $**P < 0.01$, $****P < 0.0001$

levels of IFNγ, TNFα, and the innate chemokine MIP-2, possibly secreted by the M1 macrophages[38], and decreased the levels of VEGF and IL-6 from both cancer cells and within the tumor microenvironment. VEGF is not only important in tumor angiogenesis, but also in actively suppressing the anti-tumor immune response by increasing tumor-infiltrating Tregs, TAMs, and MDSCs, while decreasing APC maturation and T cell infiltration and effector function[39]. Likewise, IL-6 can promote tumorigenesis and is linked to prostate cancer progression and MDSC recruitment[40]. Thus, through multiple mechanisms, CP1 reprogrammed the prostate tumor microenvironment, thereby sensitizing tumors to anti-PD-1 immunotherapy (Fig. 8).

In contrast to the single-armed bacillus Calmette-Guerin (BCG) and Toll-like receptor (TLR) agonists[41], CP1 can induce ICD and is capable of colonizing tumor tissue to continually enforce its multi-faceted immunomodulatory abilities. Unlike many other microbial-based therapies, CP1 is a clinically isolated patient-derived bacteria, contributing to its innate prostate-tropic properties. Further, CP1 did not cause global immune

amplification, but rather decreased immunosuppressive intratumoral Tregs and VEGF while increasing important anti-tumor immune cell types and cytokines.

Another strength of this study was in representing multiple clinically relevant genetic backgrounds of prostate cancers. We utilized the androgen-dependent Myc-CaP cell line[21], driven by *MYC* overexpression, as is seen in 80–90% of human prostate tumors[42]. Additionally, we performed a CRISPR-Cas9 knock out of *PTEN* from the Myc-CaP genome, as concurrent *MYC* copy number gain and *PTEN* copy number loss is associated with prostate cancer-specific mortality and is reported in over half of deadly metastatic prostate tumors[27]. *PTEN* loss is also linked to increased PD-L1 expression[25], decreased TILs, and resistance to PD-1 blockade[26]. Further, both cell lines were surgically injected in the prostate, allowing for orthotopic tumor development with a prostatic immune microenvironment and endogenous dLNs. In both models, a single dose of CP1 significantly augmented the anti-tumor response to significantly increase survival and

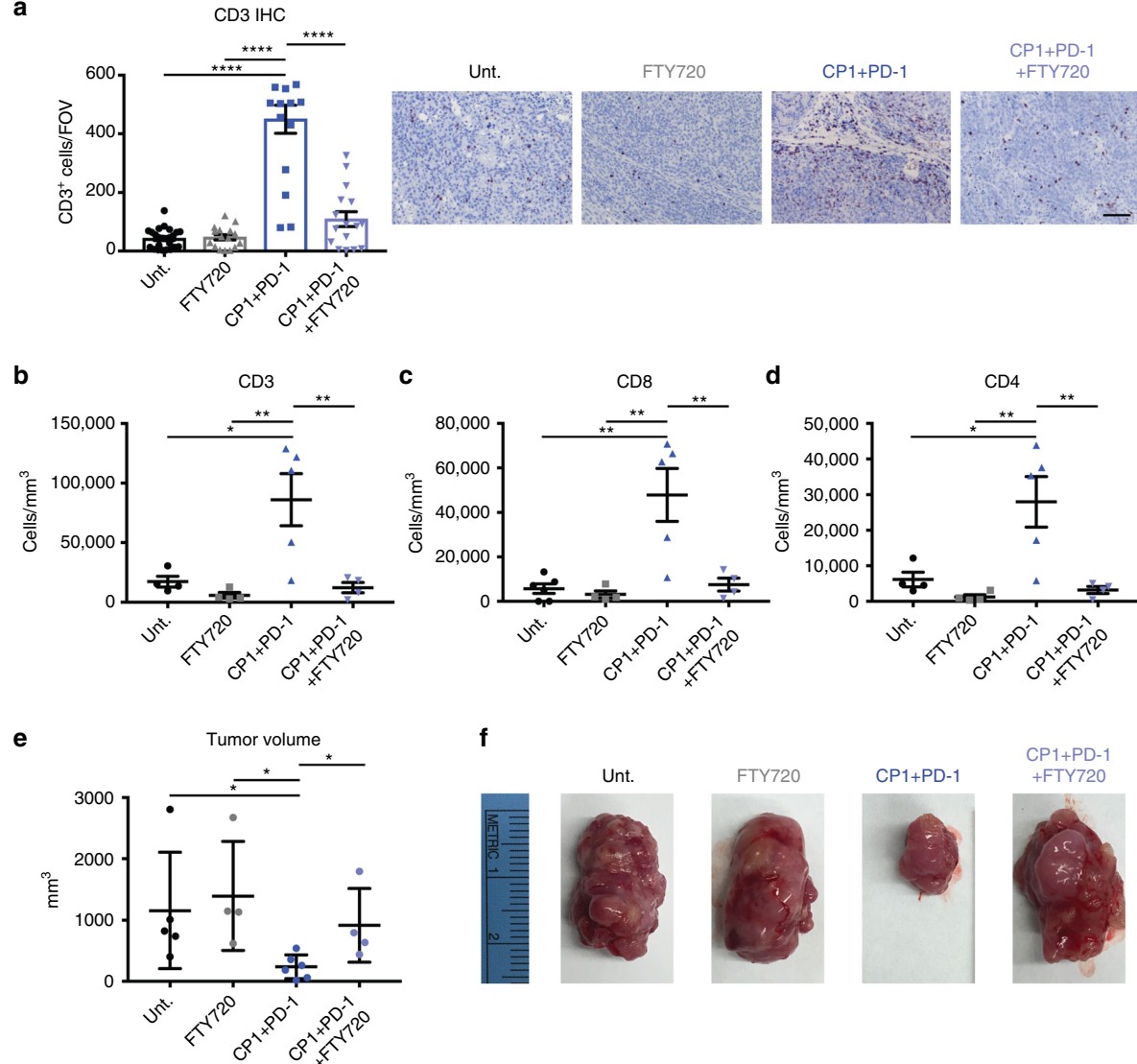

**Fig. 7** CP1 therapeutic efficacy is dependent on its recruitment of TILs. Myc-CaP PTEN KO tumor bearing mice untreated or treated with FTY720, CP1 and anti-PD-1 antibody, or CP1 and anti-PD-1 antibody and FTY720. **a** IHC with representative images, quadruplicate FOVs scored per sample (scale bar, 100 μm). **b**–**d** Flow cytometry analyses of tumors. Post-tx tumor **e** volumes and **f** gross images; $n = 4$–6 mice/experimental group. Data represented as mean ± S.E.M. Statistical significance was determined by **a**–**d** one-way ANOVA, **e** two-tailed Student's $t$-test. *$P < 0.05$, **$P < 0.01$, ****$P < 0.0001$

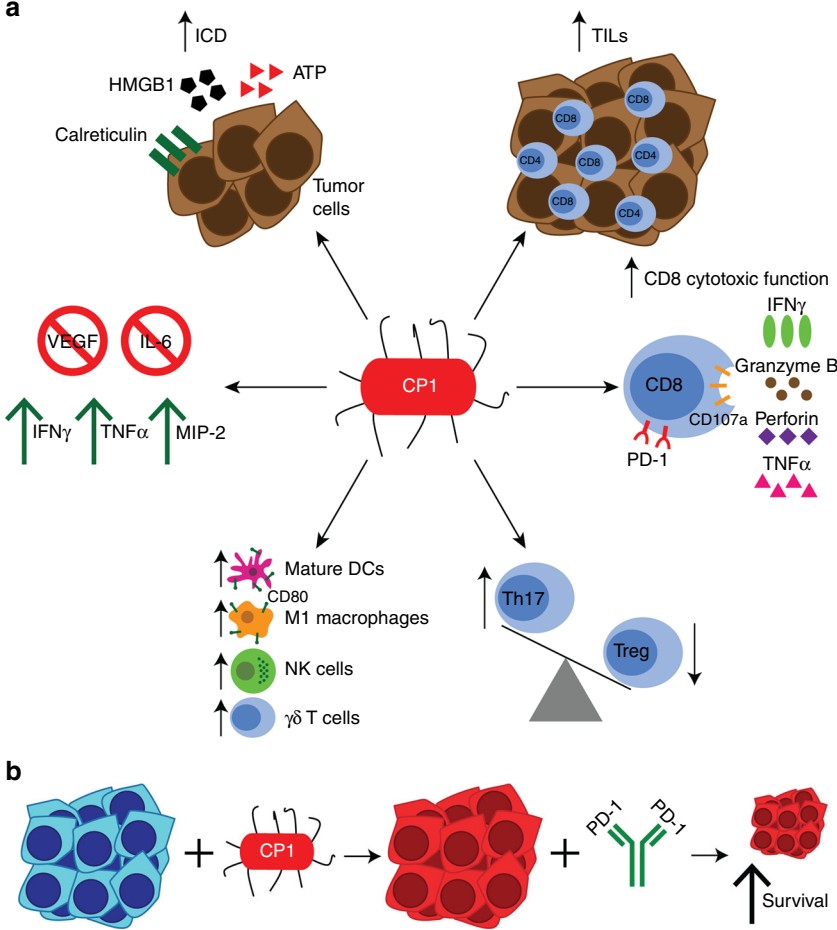

**Fig. 8** CP1 is a tissue-specific and multi-faceted immunotherapeutic tool. **a** Intra-urethrally administered CP1 colonized tumor tissue and increases CD8 and CD4 TILs, T cell cytotoxic function via IFNγ, granzyme B, perforin, and TNFα expression, skews the Th17/Treg axis to increase Th17 cells and decrease Treg TILs, increases tumor infiltration of mature DCs, M1 macrophages, NK cells, and γδ T cells, decreases intra-tumoral VEGF and IL-6 and increases IFNγ, TNFα, and MIP-2, and directly kills cancer cells with induction of immunogenic cell death (ICD). **b** CP1 reprograms non-immunogenic "cold" prostate tumor microenvironment and sensitizes tumors to anti-PD-1 blockade, resulting in decreased tumor burden and increased survival

decrease tumor burden when combined with less than 2 weeks of PD-1 blockade.

Multiple future steps can be taken to overcome limitations and achieve optimal clinical success. While we extensively evaluated the safety of CP1, future studies can attempt to attenuate the bacteria without diminishing the pro-inflammatory qualities important for its anti-tumor efficacy. To increase its therapeutic potency, CP1 can be engineered similarly to prior bacterial cancer therapies to deliver cytotoxic agents, cytokines or chemokines, tumor antigen, or genetic material[43]. Further, CP1 increased tumor infiltrating B cells, and while B cells are able to present antigen at comparable levels as mature DCs[44] and are associated with increased survival[45], others have demonstrated tumor-promoting and immunosuppressive properties of B cell subsets in prostate cancer[46]. Therefore, CP1 may be optimized with a B cell inhibitor, such as rituximab, which has previously resulted in a biochemical response in a patient with advanced prostate cancer[47].

In summary, CP1 is a single intra-urethral dose, multi-faceted immunotherapeutic tool capable of homing to prostate tumor tissue and reprogramming the "cold" tumor microenvironment, thereby sensitizing the otherwise resistant cancer to immune checkpoint inhibition. The combination of CP1 and PD-1 blockade achieved these results in multiple, clinically relevant, orthotopic models of the highly prevalent and deadly disease.

More broadly, this study demonstrates how select tissue-specific microbes, as are commonly isolated colonizing the breast[12], pharynx[13], intestines, bladder[14], female genital tract[15], and additional tissues throughout the body, can be screened and evaluated to uncover future CP1-like bacteria to potentiate immunotherapies in other recalcitrant cancers.

## Methods

**Mice.** All FVB/NJ mice (The Jackson Laboratory) used in this study were housed in a pathogen-free animal barrier facility or a containment facility, as appropriate. All experiments involving mice utilized male FVB/NJ mice administered intra-prostatic cancer cells when 6–8 weeks old, with the number of mice indicated in the respective figure legends. All experiments and procedures were performed in compliance with ethical regulations and the approval of the Northwestern University Institutional Animal Care and Use Committee (IACUC).

**Cells lines and tissue culture.** Myc-CaP, LNCaP, and 293T cells lines (ATCC) used in this study were verified to be mycoplasma-free (Biotool), and human cell lines were authenticated by short tandem repeat (STR) loci profiling (ATCC). 293T cells were grown in DMEM (Corning), Myc-CaP and LNCaP in RPMI (Gibco), all supplemented with 10% heat inactivated fetal bovine serum (FBS; Corning) and 1% Penicillin-Streptomycin (10,000 U/ml; Life Technologies). All cell culture was performed in a 37 °C 5% $CO_2$ incubator with phosphate buffered saline (PBS; VWR) and 0.25% trypsin-EDTA (Gibco).

**Bacterial growth and inoculation.** CP1 and MG1655 were grown as previously described[17]. Bacteria were grown in Luria Broth (LB) media (Sigma) at 37 °C for 24 h shaking followed by 24 h static, and were subsequently resuspended in PBS at

$2 \times 10^{10}$ cells/ml. For indicated in vitro assays, CP1 was heat killed at 70 °C for 45 min. For in vivo experiments, 10 µl of CP1 ($2 \times 10^8$ cells), MG1655 ($2 \times 10^8$ cells), or sterile PBS were administered intra-urethrally by catheterization to isoflurane anesthetized mice[48].

**Library construction and whole-genome sequencing.** Library construction and sequencing were performed at the Northwestern University sequencing core facility. DNA libraries were prepared using a Nextera XT DNA Library Preparation Kit (Illumina) per the manufacturer's instructions, and then sequenced. Briefly, a tagmentation reaction was performed which fragmented the DNA and ligated on appropriate PCR. A limited-cycle PCR reaction was then performed which added barcode and sequencing adapter sequences to the tagmented fragments. Very short fragments were removed from the PCR product using Agencourt AMPure XP beads (Beckman Coulter). The distribution of fragment sizes in the libraries was assessed on an Agilent 2100 Bioanalyzer using a High Sensitivity DNA Assay (Agilent), and the concentrations were measured using a Qubit assay (Fischer Scientific). The resulting libraries were then normalized, pooled, and loaded on to a MiSeq (Illumina), using V3 chemistry, to generate paired-end 300 bp reads.

**Whole-genome assembly and annotation.** DNA quality was assessed using FastQC, and the reads were trimmed using Trim Galore. The trimmed reads were then used to generate genome assemblies with SPAdes version 3.11.1[49] using the default parameters for paired-end reads. The resulting assembly graphs were visualized using Bandage[50]. The genome was annotated with Rapid Annotation using Subsystem Technology (RAST)[51] and analyzed and visualized using RAST and Artemis[52]. CP1 was stratified into one of the four major E. coli phylogenetic groups (A, B1, B2, or D) using the Clermont method based on chuA, yjaA, and the DNA fragment TSPE4.C2[53], and MLST was performed based on the Warwick Medical School scheme of seven housekeeping genes: adk, fumC, gyrB, icd, mdh, purA, and recA[54]. The phylogenetic tree was created with concatenated MLST sequences of CP1 and reference E. coli strains using the Maximum Likelihood method with MEGA7[55].

**Gentamicin protection assay.** As previously described[17], tumor cells were incubated with CP1 or MG1655 (MOI 1) in antibiotic-free media for 2 h at 37 °C 5% $CO_2$. To quantify bacterial invasion, cells were washed four times with PBS, treated with 50 µg/ml gentamicin, incubated with 0.05% trypsin/0.1% Triton X-100 for 10 min at 37 °C 5% $CO_2$, and cells were harvested, plated on LB agar, and colonies counted after 24 h. To quantify bacterial adherence, cells were washed, immediately incubated in trypsin/Triton X-100, collected and plated, and adherence was measured as the difference from invasion colony counts. To quantify intracellular proliferation of bacteria, the cells were washed and incubated with 50 µg/ml gentamicin for 22 h at 37 °C 5% $CO_2$, followed by cell collection.

**ICD and cell death assays.** Cell death from CP1 or MG1655 (MOI 1 or 10) and cancer cell co-culture was measured by supernatant lactate dehydrogenase (LDH; Cytotoxic 96 Non-Radioactive Cytotoxicity Assay, Promega). For in vitro ICD assays, 1 µM mitoxantrone was used as a positive control. Supernatants were collected and cell counts performed after 72 h for quantifying secreted ATP (Bioluminescent Assay Kit, Sigma) and high mobility group protein B1 (HMGB1; ELISA, Tecan Trading). Also after 24 or 72 h, cells were incubated with rabbit anti-calreticulin (Abcam ab2907 1:1000) for 60 min, followed by Alexa Fluor 488 anti-rabbit secondary (Invitrogen A11008 1 µg/ml) for 30 min, and analyzed by flow cytometry. For in vivo ICD assays, HMGB1 immunofluorescence (IF) was quantified as the percentage of HMGB1$^-$ nuclei and calreticulin IF was analyzed for cell surface staining. In vitro, caspase 3/7 activity was assessed at 6 or 24 h (Caspase-Glo 3/7 Assay, Promega). Early (Annexin V$^+$ PI$^-$) and late (Annexin V$^+$ PI$^+$) stage apoptosis were analyzed at 24 h by flow cytometry (Annexin V Apoptosis Detection Kit, eBioscience). Phosphorylated MLKL, MLKL, RIP1, and full length and cleaved PARP were analyzed by Western blot. As indicated, select experiments included the addition of 50 µg/ml gentamicin 2 h after co-culture initiation.

**Multiplex cytokine/chemokine array.** Tissue lysates were prepared in RIPA buffer (Sigma) supplemented with protease (cOmplete Tablets, Mini, EDTA-free, Roche) and phosphatase (PhosSTOP, Roche) inhibitors. Tissues were homogenized using an electric pestle or a gentleMACS dissociator in M Tubes (MACS Miltenyi Biotec). Protein from tissue (10 µg) or in vitro supernatant (25 µl) was added per well of a 32-plex mouse cytokine/chemokine magnetic bead milliplex plate (EMD Millipore), which was run using a MAGPIX Luminex plate reader (Thermo Fisher Scientific) and analyzed on xPONENT Software Solutions.

**293T transfection and lentiviral transduction of tumor cells.** Lentivirus was produced by co-transfection of 293T cells with 3 µg luciferase expressing vector pLV-mCherry-P2A-luciferase, 2 µg Δ8.9 HIV-1 packaging vector, 1 µg VSVG envelope glycoprotein vector, and 2.5 µl/µg Lipofectamine 2000 (Invitrogen) in Opti-MEM media (Gibco) in 6-well plates at 37 °C 5% $CO_2$ for 16 h. Supernatant virus was collected, 0.45 µm filtered, and diluted 1:5 and supplemented with 8 µg/ml polybrene (Santa Cruz Biotechnology) before spinfecting Myc-CaP cells for 2 h

at 32 °C. At least 48 h later, mCherry positivity was verified and sorted for top 10% positivity using a FacsAria SORP cell sorter (BD).

**Orthotopic surgical tumor model and treatment regimens.** Intra-prostatic surgical injections were performed as previously described[56] with modifications. Mice were administered at least 0.05 mg/kg pre-operative buprenorphine and anesthetized with isoflurane, verified by toe pinch. The abdominal region was shaved and sterilized, and $1 \times 10^6$ tumor cells in 30 µl (1:1 PBS and matrigel [Basement Membrane Mix, Phenol Red-Free, LDEV-Free, Corning]) was injected (Hamilton syringe and 28-gauge needles) into one anterior prostate lobe, initially verified by engorgement of the lobe. The inner abdominal wall was closed with 5–0 absorbable sutures (J493G, eSutures) and the outer skin was closed with 4–0 non-absorbable sutures (699H, eSutures). Approximately 1 mg/kg post-operative meloxicam was administered immediately, 24 and 48 h post-surgery. Sample size was determined with consideration of the duration of each intra-prostatic surgery and the power needed to allow for adequate statistical analyses with the number of experimental groups. Survival endpoint was defined in advance as the appearance of hemorrhagic abdominal ascites[57] and/or decreased grooming, ambulation, or piloerection[58]. Tumor volumes were calculated using caliper measurements at $\pi/6 \times L \times W \times H$, where $L$ was the length of the longest axis of the tumor, and $W$ and $H$ were the perpendicular width and height, respectively. CP1 was administered intra-urethrally to tumor-bearing mice on day 8 post-tumor injection. One-hundred micrograms of anti-PD-1 antibody (RMP1-14, BioXCell) or IgG2a isotype control (2A3, BioXCell) were administered intra-peritoneally (i.p.) every other day, as previously performed[59], from day 17–29 for Myc-CaP and from day 13–23 for Myc-CaP PTEN KO experiments. For select experiments, FTY720 (Sigma) was administered 25 µg intravenously (i.v.) 24 h prior to CP1 administration followed by 5 µg i.p. daily until analysis, as previously performed[60]. Mice were not randomized, rather pre-treatment tumor imaging was utilized to normalize tumor burden and variance among all experimental groups before CP1 or anti-PD-1 antibody administration (Supplementary Figure 8). Blinding during the course of treatment was not possible to prevent cross-contamination between CP1-infected and non-infected mice during handling and the daily (FTY720) and/or every other day (anti-PD-1 or isotype antibody) injections. Investigators were blinded to some outcome analyses.

**In vivo bioluminescent imaging.** Luciferase-expressing tumor-bearing mice were injected i.p. with 10 µl/g body weight of 15 mg/ml 0.22 µm filtered D-luciferin (sodium salt, Gold Bio). At least 10 min after injection, the mice were imaged with an IVIS Spectrum Imaging System (PerkinElmer). Images were analyzed and quantified using Living Image software.

**In vivo bacterial colonization.** Mice tissues were analyzed at day 1 or day 9 after intra-urethral CP1 administration. As previously described[17], tumors, bladders, kidneys, livers, and spleens were aseptically excised, dissected, homogenized by electric pestle, and plated in serial dilutions on eosin methylene blue (EMB) agar and incubated at 37 °C for 24 h.

**RNA extraction and qRT-PCR.** Excised tissue was immediately placed in RNA later until homogenization using TissueMiser Homogenizer (Fisher Scientific) or gentleMACS Dissociator in M Tubes (MACS Miltenyi Biotec). RNA was extracted by Trizol (Thermo Fisher Scientific) and subsequent RNAeasy Plus Mini kit (QIAGEN), and complementary DNA (cDNA) was generated using oligo d(T)$_{16}$ primer (Invitrogen) and random hexamer (Promega) at 65 °C for 5 min, followed by the addition of dNTPs (Promega), 1× first strand buffer (Invitrogen), DTT (Invitrogen), SUPERase-In RNase inhibitor (Invitrogen), and M-MLV reverse transcriptase (Invitrogen) at 25 °C for 10 min, 37 °C for 50 min, and 70 °C for 15 min. Quantitative reverse transcription-PCR (qRT-PCR) was performed using a QuantStudio 6 Flex Real-Time PCR System (Applied Biosystems) at 50 °C for 2 min, 95 °C for 10 min, and 40 cycles of 95 °C for 15 s, 60 °C for 15 s, and 72 °C for 1 min using SYBR Green master mix (Bio-Rad) and the following primers: 16S (F: ACTCCTACGGGAGGCAGCAGT, R: TATTACCGCGGCTGCTGGC) or the mouse housekeeping gene RPLP0 (F: AGATGCAGCAGATCCGCA, R: GTTCTTGCCCATCAGCACC) (Integrated DNA Technologies). Data were analyzed using QuantStudio Real-Time PCR software. In Supplementary Figure 3d, 16S qRT-PCR results from CP1-administered tumors were calibrated to 16S values of CP1 titrations of known cell counts ($R^2 = 0.98184$), and were subsequently normalized to calibrated PBS-administered tumors, total RNA yield, and the weight of the tumor tissue from which RNA was extracted.

**Flow cytometry.** Single-cell suspensions were generated from tumors using a gentleMACS Dissociator with Heaters with the Tumor Dissociation Kit in C Tubes (MACS Miltenyi Biotec). Tissues were passed through a 70-µm filter, resuspended in 30% Percoll (Sigma), and overlayed on top of 70% Percoll, centrifuged without brakes, and the buffy coat layer was isolated and viable cells counted. Tumor-draining para-aortic lymph nodes single cell suspensions were created by passing cells directly through a 70-µm filter, followed by red blood cell lysis with ACK buffer (0.15 M NH$_4$Cl, 10 mM KHCO$_3$, 0.1 mM Na$_2$-EDTA; pH 7.2–7.4; 0.2 µm filtered). All samples were treated with anti-mouse CD16/CD32 Fc block (2.4G2,

BD). For intracellular staining, cells were resuspended in RPMI 10% FBS with 50 ng/ml PMA (Sigma), 1 μg/ml ionomycin (Cell Signaling), 1 μl/ml brefeldin A (GolgiPlug; BD), 2 μl/3 ml monensin (GolgiStop; BD), and CD107a antibody when appropriate, for 6 h at 37 °C 5% $CO_2$. Antibodies utilized for flow cytometry are listed in Supplementary Table 1, and all antibodies were individually titrated to determine optimal staining dilutions. After subsequent extracellular staining, cells were stained with LIVE/DEAD Fixable Blue Dead Cell Stain Kit (Invitrogen). FoxP3 panels were fixed and permeabilized with the FoxP3/Transcription Factor Staining Buffer Set Kit (eBioscience) before antibody incubation. All other panels were fixed in IC fixation buffer (eBioscience) before subsequent permeabilization with the Intracellular Fixation and Permeabilization Buffer Set Kit (eBioscience) and incubation with intracellular antibodies when appropriate. Samples were run on a LSRFortessa 6-Laser (BD). Controls and compensation were performed using anti-rat/hamster Ig, κ/negative control compensation particles set (BD) and appropriate fluorescence minus one and unstained controls. Data were analyzed using FlowJo software. A representative flow cytometry gating strategy is displayed in Supplementary Figure 13 (a: tumor, b: dLNs), with initial gating on overall morphology, singlets, live cells, and CD45 positivity before proceeding with all further analyses.

**Histology.** Tissues were fixed in 10% neutral buffered formalin for 24–48 h at 4 °C before paraffin processing at the Northwestern University histology core. For immunohistochemistry (IHC), 5 μm sections were deparaffinized and rehydrated, followed by antigen retrieval with citrate buffer pH6 (Dako) or 1 mM EDTA pH 8, 3% $H_2O_2$ (Sigma), blocking (BioCare Blocking Reagent BS966M, Dako X0909, or Vector ImmPRESS 2.5% normal horse serum), primary antibody incubation, secondary antibody incubation (Vector biotinylated anti-rat IgG, Dako EnVision+ System HRP, Vector ImmPRESS HRP), streptavidin-HRP (Biocare) when appropriate, 3,3′-Diaminobenzidine chromogenic detection (SIGMA*FAST* tablets, Sigma), hematoxylin counterstain (Vector), tissue dehydration, and slide mounting (Cytoseal-XYL). IHC (Fig. 3) slides were blinded and scored manually over the entire tissue surface area or were quantified using ImageJ with quadruplicate field of views (FOVs) analyzed per sample. For *E. coli* IF, as previously described[61], slides were deparaffinized and rehydrated, antigen retrieval was performed with citrate buffer pH6 (Dako), and slides were blocked with 10% goat serum (Vector Laboratories). Slides were then incubated with primary antibody, then streptavidin-Alexa Fluor 594 secondary antibody (ThermoFisher Scientific, 1:500), followed by permeabilization with 0.25% Triton X-100, repeat primary antibody, anti-rabbit IgG (H + L) Alexa Fluor 488 secondary antibody (ThermoFisher Scientific, 1:500), and DAPI (Sigma) counterstain, and were subsequently mounted with ProLong Gold Antifade Mountant (Molecular Probes), resulting in green intracellular staining and red/yellow (green + red) extracellular staining. HMGB1 and calreticulin immunofluorescence was similarly performed with Alexa Fluor 488 secondary antibody. Primary IHC and IF antibodies are listed in Supplementary Table 2. Brightfield images were taken with a SPOT RT Color camera on a Olympus CKX41 inverted microscope and IHC and IF images with CRI Nuance spectral camera on a Zeiss Axioskop upright microscope or a NikonDS-Ri2 microscope.

**Chemistry panel and complete blood count.** Mouse peripheral blood was collected by cardiac puncture and placed in serum separator or dipotassium-EDTA tubes (BD Microtainer). Frozen serum and whole blood were analyzed, the latter within 24 h after collection (Charles River Laboratory). Reference value ranges were used from the Charles River Laboratory[62], the University of Arizona University Animal Care (https://uac.arizona.edu/clinical-pathology), and the University of Minnesota Research Animal Resources (http://www.ahc.umn.edu/rar/refvalues.html).

**CRISPR knockout.** To stably express CAS9 in Myc-CaP cells, we generated VSVG pseudotyped lentivirus[63,64] using 293T cells, 2nd generation packaging vectors psPAX2, pMD2.G, and a CAS9 (*Streptococcus pyogenes* CRISPR-Cas) expressing lentiviral vector (Addgene 52962)[65]. Lentiviral infection efficacy was >90% and cells were maintained with 8 μg/ml puromycin. Multiple synthetic guide RNAs (gRNAs) (CRISPR crRNA, Integrated DNA Technologies) were designed using the CRISPR Design Tool (crispr.mit.edu[66]), those with off-target effects were excluded. gRNAs were delivered by transient transfection reagent TransIT-X2 (Mirus Bio). Partial PTEN knockout was confirmed by PTEN and p-AKT western blot and IF; >40 clones were isolated by cloning cylinders and were screened for complete PTEN loss. Two complete PTEN knockout Myc-CaP clones from different gRNAs (AAAGACTTGAAGGTGTATAC (exon 2), TGTGCATATTTATTGCATCG (exon 5)) were selected and analyzed in parallel in vitro.

**Cancer genomic database analysis.** cBioPortal for Cancer Genomics (http://www.cbioportal.org)[67] was utilized to analyze The Cancer Genome Atlas (TCGA) Research Network (http://cancergenome.nih.gov/)[68] and the Stand Up To Cancer/Prostate Cancer Foundation (SU2C/PCF) database[69].

**Western blot.** Western blotting was carried out as previously described[70]. Overall, lysates were prepared in RIPA buffer (Sigma) supplemented with protease (cOmplete Tablets, Mini, EDTA-free, Roche) and phosphatase (PhosSTOP, Roche) inhibitors. Protein lysates were loaded with 4× Laemmli sample buffer (Bio-Rad) and 2-mercaptoethanol (Bio-Rad) and run by SDS-PAGE, and were subsequently transferred to PVDF membranes (Bio-Rad) using Trans-Blot Turbo transfer buffer (Bio-Rad) and a Trans-Blot Turbo Transfer System (Bio-Rad). Membranes were blocked for 1 h at room temperature with 5% blotting-grade blocker non-fat dry milk (Bio-Rad), followed by overnight 4 °C incubation with the appropriate primary antibody (Supplementary Table 3), and 1 h room temperature incubation with an anti-rabbit or anti-mouse IgG (H + L)-HRP conjugate (Bio-Rad) secondary antibody. Blots were imaged using chemiluminescent substrate (Thermo Scientific) and a LAS-3000 imager (Fujifilm) or ChemiDoc Imaging System (Bio-Rad). When necessary, blots were stripped with Restore PLUS western blot stripping buffer (Thermo Scientific), followed by repeat blocking and antibody incubations. Uncropped images of the western blots from Fig. 5a are presented in Supplementary Figure 14.

**Cell proliferation assay.** Cell proliferation was assessed by quantification of MTS tetrazolium reduction (Promega). Select experiments were performed with low (1%) or charcoal-stripped (C.S.) FBS.

**Organoid culture.** Cells were resuspended in Hepatocyte Defined Medium (Corning) supplemented with 10 ng/ml epidermal growth factor (Corning), 5% C. S. FBS, 1× Glutamax (Gibco), 5% matrigel (Corning), 10 μM ROCK inhibitor (Y-27632, STEMCELL Technologies), 100 nM DHT (Sigma), and 1× Gentamicin/Amphotericin (Lonza). Cells were plated in Ultra-Low Attachment Surface plates (Corning).

**Statistical analyses.** Statistical analyses were performed in GraphPad Prism 7 software. The number of technical replicates, biological replicates, and independent experiments performed are listed in the figure legends. Unpaired two-tailed Student's *t*-test, one-way Analysis of Variance (ANOVA) with post-hoc Tukey, and two-way ANOVA with post-hoc Sidak were utilized as appropriate. Survival studies were analyzed by Log-rank (Mantel-Cox) test. Correlations were analyzed by Pearson's correlation coefficient (*r*). Slopes of linear regression trend lines were compared by Analysis of Covariance (ANCOVA). Data are presented as mean ± standard error of the mean (S.E.M.), unless otherwise indicated. All data were included, no outliers were excluded. For all analyses, results were considered statistically significant with $P < 0.05$, *$P < 0.05$, **$P < 0.01$, ***$P < 0.001$, and ****$P < 0.0001$.

**Data availability.** All data of this study are available within this article and its Supplementary Information Files or are available from the corresponding authors upon reasonable request. The CP1 Whole Genome Shotgun project has been deposited at DDBJ/ENA/GenBank under the accession PZKJ00000000. The version described in this paperis version PZKJ00000000.

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

## Acknowledgements

We thank the members of the P.T. and S.A.A. laboratories, B. Zhang, M. Brown, and D. Wainwright for their support, the DNA/RNA Delivery Core of the Skin Disease Research Center of Northwestern University for their assistance with CRISPR, S. Swaminathan and the Robert H. Lurie Comprehensive Cancer Center Flow Cytometry Core for help with designing the flow cytometry staining panels, M. Sebastian and the MD Anderson Cancer Center Histology, Pathology, and Imaging Core (supported by P30 CA16672 DHHS/NCI Cancer Center Support Grant) for blinded IHC staining and scoring, S. Murphy for bacterial genomic DNA preparation, and B. Wray and the NUSeq Core facility at Northwestern University for assistance with bacterial whole-genome sequencing. This work was supported by National Institutes of Health grants to J.F.A. (NCI F30 CA203472), S.A.A. (NCI R01 CA167966, NCI R01 CA123484, NCI P50 CA180995), and P.T. (NIDDK R01 DK094898, NIDDK R01 DK108127).

## Author contributions

J.F.A., S.A.A., and P.T. conceived the project, designed the research studies, analyzed the data, and provided manuscript review. J.F.A. conducted all the experiments, data acquisition, and wrote the manuscript. A.F.N. and H.M. aided in orthotopic surgeries. A. J.S. provided experimental feedback and manuscript review.

## Additional information

**Competing interests:** J.F.A., P.T., S.A.A., and A.J.S. are co-inventors on a provisional patent on CP1 use through Northwestern University on "Immunostimulatory bacteria for the treatment of cancer" under filing with the US Patent office (Application number 62539843). The remaining authors declare no competing interests.

