## [Peer Review File · Nature Communications]

Reviewers' comments:

Reviewer #1 (Remarks to the Author):

In this manuscript, the authors describe a series of nicely done studies using an E Coli strain (CPI) that leads to chronic infection in humans. This strain (but not a control strain) appears to home to prostate tumor tissue and in vitro it mediates lysis of prostate tumors. This is an immunogenic cell death, and accompanied by the expected up-regulation of HMGB-1, calreticulin and ATP. Immunologically, trans-urethral administration of CPI leads to up-regulation of IFN-gamma expressing CD8 in the prostate tumors, an increase in the CD8 / Treg ratio and also an increase in TH17 which is surprising given that CP1 causes a decrease in IL-6 which is of course critical for TH17 induction. Anti-tumor effects of CPI = PD-1 blockade are documented in both a myc-driven and a myc / pten KO cell line (a new and novel reagent). Overall the studies are important, well-controlled and the conclusions drawn are well-supported by the data provided. I have only minor comments:

MINOR:

- 1) It's not particularly clear why MG1655 is a good control here, this should be more adequately explained in the results section.
- 2) The direct lysis of tumor cells by CPI is interesting, do the authors have any data regarding the mechanism of this lysis?
- 3) One important tissue type missing from Figure 1C is normal prostate epithelium, one would assume that CPI would at least adhere to that tissue. Data on normal prostate tissue really should be included in Figure 1C if they're available.
- 4) Line 108 – the authors should denote how they defined MDSC in both the results section and in the figure legend, this is important because there are many competing definitions. I realize this is in the methods, it should be repeated here for clarity.
- 5) Line 102- Given that the CD8 T cells that infiltrate the tumor express IFN-gamma, PD-1 is certainly not an exhaustion marker here. PD-1 increases with T cell activation, so in this setting it's simply an activation marker.
- 6) Line 150 – the Myc / PTEN loss model is interesting, do those cell metastasize in vivo?
- 7) In the abstract the authors state that TH17 cells mediate lysis of tumor cells, but this is not documented in the results section as far as I could see. Direct lysis of tumor cells by CD4 T cells is not without precedent, but if the authors wish to claim that their TH17 are mediating lysis additional studies are required. Specifically, they'd need to isolate (sort) TH17 from tumors (using a bead capture method) and perform an in vitro CTL assay. While that would be an interesting experiment, it's really not required or relevant to the present study. Instead, it might be more facile to simply edit the abstract etc.

Reviewer #2 (Remarks to the Author):

In this manuscript, it is demonstrated that the UPEC strain CP1 can accumulate in tumors in the prostate when delivered intra-urethrally. CP1 appears to work synergistically with anti-PD-1 to delay progression by recruiting parts of the immune system. While the numbers of animals are too few in some experiments, this conclusion appears to be likely and is important.

The manuscript has a large number of rectifiable deficiencies. Vital information about the experimental design in many experiments was missing or sufficiently hard to find that I missed it. Claims were unnecessarily exaggerated in places. Trying too hard to impress made some very simple things harder to understand. I persisted because a manuscript that "cut to the chase" would have merit.

A sufficient number of bacteria get to the tumor and persist there to allow a useful immunological reaction. However, what is that number? It was impossible to tell. Near the beginning of colonization, it appears to be 2% of the inoculum can be plated out of the tumor. The 16S qPCR

used later needs to be calibrated so can know if there are hundreds or millions of bacteria at later time points. What proportion are intracellular? What proportion are still capable of replication? Perhaps there are all persisters or VBNC. Under many scenarios, it is implausible that ICD is happening on a significant scale, and the pertaining parts could be deleted without any loss of significance of the observation.

If the tumors in the CP1 mice were smaller at the time of sacrifice due to sickness, what caused the mice to be sick? Did metastases occur? If the bacteria did not disseminate, metastases might grow freely.

CP1 are localized. This means metastases are not targeted. Is that a good thing or a bad thing? I would be curious as to how intra-tumor LPS or K12 would behave in this model to dissect out exactly what is necessary for the observations. What happens if you treat with an antibiotic to remove the CP1 after a short time?

Below I go through the manuscript, line by line.

Lines 64 and 65. Not unique. This is novel but likely not unique. It is not even surprising, in retrospect, given that UPEC invade epithelial cells. It is still important. No need to stretch. Just the facts.

Line 69. The data regarding virulence factors is based on negative PCR results from other papers. Given the work done on this bacterium, it would be a good idea to spend \$300 on sequencing the genome.

Line 70: It is claimed that the specificity for prostate cancer cells is being tested. This is an exaggeration. No evidence is provided that CP1 is any different from any other UPEC in its abilities. Indeed, its genetics may mean it is "worse" than many other UPEC, which could be a good thing for a living therapeutic.

Line 73: Fig 1a. The laboratory K12 strain does not adhere well, does not invade, and does not proliferate intracellularly. UPEC do. This is a straw man. Supplement.

Line 74: It is not clear if Fig. 1b was done in Gentamycin or not, and for how long. I can comment when I know. Supplement.

Line 79. The timing and units really matter because overall, it is not clear how much proliferation is taking place, if any. I am guessing this was about two days after administration, at which point bacteria representing 2% of the inoculum were found in the tumor.

In summary for this part of Figure 1:

"UPEC colonize the urinary tract and are known to invade and proliferate in epithelial cells (REFS). CP1 was isolated from a prostate and is in a rare UPEC phylogenetic group that lacks many of the virulence factors found in other phylogenetic groups (REF), so it is potentially less virulent. We demonstrated that the ability to invade epithelial cells extends to a mouse epithelial prostate cancer cell line, at least in the case of CP1. (supp figure 1a and b). It has been established that an intraurethral dose of $\approx 10^8$ UPEC bacteria, including CP1 (REF), leads to colonization of the bladder and prostate without progressing to the kidney or progressing to the systemic tissues. Using this delivery model, we reiterated these results and additionally showed that CP1 can be isolated from an orthotopically implanted syngeneic prostate tumor. Delivery was X days after implantation and recovery was Y days later and over 10^6 bacteria per (what units?) were recovered (grams? Tissue, what?) (Figure 1c, now a)."

The whole of the results need the same streamlining and clarity.

Line 80. There is no hint as to why Fig. 1d shifts to a 16S PCR assay from CFUs. Explain why. Here and elsewhere, there is a need to calibrate the number of bacteria with the reference gene and with the number of host cells in the sample. We will learn later that tumors vary a lot in size, partly depending on the treatment. Without knowing the size of the sample (and a calibration to approximate CFU) we have no idea how many bacteria are present and whether variation is simply because non-replicating bacteria are diluted to different degrees.

Line 81. In fig. 1e, we see some green and yellow spots. What is the proportion inside and out? Is there an estimate of how many bacteria are present? How long after the tumor was formed did the delivery of bacteria take place? How long after delivery was the imaging was performed? Is there any evidence that the bacteria divide after they arrive at the prostate?

Line 90. It is hard to interpret figure 2 reliably without knowing whether this involved a

gentamycin protocol or not. What were the number of bacteria inside and outside the host cell at 72hrs? If there was no gentamycin then one has to imagine that 10^{10} cfu/ml of a control K12 bacterium after 72hrs might have a similar adverse effect.

Fitting this experiment into the subsequent theories is dependent on how many bacteria are present at 72 hours in this experiment and how many are present within the tumor and replicating in the in vivo experiment.

Line 101. Fig. 3. It is not clear to me how long after tumors were introduced the bacteria were delivered, and how long after that delivery these immunophenotyping measurements were performed. How was this time chosen? Surely, it matters. Presumably, there is a time component to this response. Was this explored?

Line 125. It would be good to know the actual p value for the the KM plot.

Line 129. I presume these experiments are done at the time of sickness-induced sacrifice. What happened to mice that never became sick during the experiment?

IVIS was done before treatment. It was then done again but it is not clear when that was.

Presumably not at the time of death when the tumor can be measured directly?

Line 132. Supplementary fig. 4. These are ratios, not levels. The tumors could be of various sizes and perhaps smaller in animals that survive longer. To be careful, it would be interesting to normalize for sample volume or sample weight and plot again for the dose of CP1 present per mouse at the time of sacrifice.

It is a pity there is no experiment with another bacteria, such as Nissle, or different doses, or an antibiotic to clear bacteria earlier, or an IV delivery, or LPS delivered intra-tumor, to try to demonstrate that this particular route of delivery or this particular bacterium is of particular merit.

Line 154. When the word "significantly" is used, then a P value is desirable, especially as only the range is given in the figures. This experiment has marginal p values and there are only three samples in each treatment group in 5f, when seven were theoretically available. What happened to the one mouse in the critical CP1+PD-1 group in the KM plot?

Line 222. There is no direct evidence that there are enough actively dividing bacteria in the tumor to cause this. The in vitro experiments may contain many orders of magnitude more bacteria per host cell than the tumors, until it is demonstrated otherwise. The data may exist but I cannot see it.

Line 224. There is no necessity for these bacteria to be delivered IV. It is done to seek out metastases. Maybe someone should try to delivery these other bacteria intra-urethrally. Indeed, the fact that CP1 does not appear to seek out metastases could be construed as a weakness.

Line 225. Surely genetic manipulations in Enterobacteria are straightforward, and you later propose to do precisely that.

Line 225. Not consistently (there is overlap) but significantly.

Line 285. The papers say 10^8 and here it seems to be 2×10^8 . Clarify.

Line 286. It costs less than \$300 to sequence the genome.

Line 288. How many tumor cells?

Line 298: Typo. CPI should read CP1.

Line 769. What are the number of living replicating bacterial cells at any time in the tumor? Is this number sufficient to demonstrate that the bacteria cause cell death in the tumor, or is it merely the presence of low levels of an extremely immunogenic bacterium that is the trigger?

Reviewer #3 (Remarks to the Author):

Using two orthotopic prostatic adenocarcinomas (Myc driven or PTEN deficient and Myc driven) inoculated by intra-urethral administration, the authors demonstrate that i) the combination of a live B1 phylogenetic group uropathogenic E. coli (UPEC) called « CP1 » synergizes with anti-PD1 Ab to reduce tumor growth and increase survival, ii) to reprogramm the TME (influx of TH17 and Tc1 TIL, decrease local IL-6/VEGF, increase CXCL10, immunogenic cell death), iii) by enhancing the recruitment from draining LN (inhibition of lymphocyte egress from dLN using FTY720 blunts

the TIL accumulation and survival benefit).

This is an interesting study that deserves a particular attention. However, additional controls are missing from this first version that need to be brought up to make it more attractive.

Major comments :

1/ Potential experimental flaws : 1A/For all figures where TIL phenotypes are shown, some assays are depicted in %, others are in cells/mm³, while tumor sizes are not the same inbetween groups (timing of FACS at sacrifice). One needs to represent all cell types within tumors of quite similar volume, at early stages of treatment start, to avoid any flaws related to tumor volume (of course, T cells are sparse in a huge tumor, even when this one has responded earlier one). Fig. 7 may be subjected to this specific criticism. 1B/ in most fig. legends, the authors failed to announce how many experiments were run, how many mice/group, while some graphs or bars depict few dots...(except Fig 4).

2/ Since many prostatic cancers result from chronic prostatitis, why is the tumor microenvironment so « cold » in most of them? does it mean that the spectrum of bacteria naturally invading prostate tumors is immunosuppressive ? or that prostate tumors are not normally invaded by bacteria ? or microbes. Authors should cultivate a couple of human primary prostate cancers arising from chronic inflammation or at least performed a FISH analysis of prostate cancer tissues to visualize bacterial products in situ, as already reported for pancreatic cancers (Geller, L. T. et al. Science 357, 1156–1160 (2017)).

3/ Specificity : it is intriguing to observe that CP1 is effective at i) homing only at prostate cancer tissues and not kidney, ii) at promoting ICD, iii) ideally reprogramming the TME, iv) synergizing with anti-PD1 Ab, with no toxicity. Given this outstanding constellation, one would like to appreciate that any random bacterium would not perform as well and that these properties are somehow quite « unique ».

A « negative Gram negative control bacterium» as well as other E. coli clones or isolates should be utilized for all experiments or at least in the Myc driven tumor model.

4/ Dose effects : it would be elegant to show a dose-effect of CP1 together with a constant dosing of anti-PD1 Ab.

5/ Th2 microenvironments : from the cytokine arrays, one can see IL-5 and IL-9 dominating the scenario. This atmosphere could favor the differentiation of authentic TH9 cells. Can IL-21 blockade antagonize the synergistic effects ? is PU-1 (pathognomic of TH9 cells) overexpress in those prostatic TILs ?

Minor comment :

It is surprising that E. coli, which is not expected to be an intracellular bacterium, is cytopathogenic (LDH release). Can the authors use a battery of assays that would be convincing that « typical or atypical » cell death is occurring, such as caspase 3 or 8 cleavage, annexinV/PI stainings, PARP cleavage, RIPK1-3 activation, MLKL phosphorylation...

Response to Reviewers' comments:

Reviewer #1 Remarks to the Author):

In this manuscript, the authors describe a series of nicely done studies using an E Coli strain (CPI) that leads to chronic infection in humans. This strain (but not a control strain) appears to home to prostate tumor tissue and in vitro it mediates lysis of prostate tumors. This is an immunogenic cell death, and accompanied by the expected up-regulation of HMGB-1, calreticulin and ATP. Immunologically, trans-urethral administration of CPI leads to up-regulation of IFN-gamma expressing CD8 in the prostate tumors, an increase in the CD8 / Treg ratio and also an increase in TH17 which is surprising given that CP1 causes a decrease in IL-6 which is of course critical for TH17 induction. Anti-tumor effects of CPI = PD-1 blockade are documented in both a myc-driven and a myc / pten KO cell line (a new and novel reagent). Overall the studies are important, well-controlled and the conclusions drawn are well-supported by the data provided. I have only minor comments:

Major comments:

None

Minor comments:

1) It's not particularly clear why MG1655 is a good control here, this should be more adequately explained in the results section.

We chose MG1655 because it is a commonly utilized *E. coli* control as it is the prototypical strain from the original patient-derived K-12 *E. coli*, it has been maintained with “minimal genetic manipulation”, and its genome has been sequenced and comprehensively categorized¹. We have updated the results with this rationale (lines 83-85).

2) The direct lysis of tumor cells by CPI is interesting, do the authors have any data regarding the mechanism of this lysis?

In this manuscript we identify that immunogenic cell death is strongly induced in tumor cells after exposure to CPI (Fig. 2a-b, new Fig. 2c-d, new Supplementary Fig. 5b-c). Our further analysis has demonstrated that CP1 can also induce caspase 3/7 activity (new Supplementary Fig. 5d-f) and Annexin V surface levels (new Supplementary Fig. 5g-h), but not markers of necroptosis (new Supplementary Fig. 5i) (lines 140-150).

3) One important tissue type missing from Figure 1C is normal prostate epithelium, one would assume that CPI would at least adhere to that tissue. Data on normal prostate tissue really should be included in Figure 1C if they're available.

In this model, it was not possible to accurately and consistently grossly separate the malignant prostate tumor mass from the normal prostate tissue. However, a previous publication² has performed this analysis and demonstrated both the ability of CP1 to adhere to, invade, and intracellularly proliferate within benign prostate epithelial cell lines *in vitro* and its ability to colonize benign prostate tissue after intra-urethral administration *in vivo*. This has been noted in the text of the Results section (lines 79-81, 91-94).

4) Line 108 – the authors should denote how they defined MDSC in both the results section and in the figure legend, this is important because there are many competing definitions. I realize this is in the methods, it should be repeated here for clarity.

MDSCs were defined as CD45⁺CD11b⁺Gr-1⁺ in the flow cytometry analysis in Figure 3. This has been added to both the results section and in the figure legend (line 170, 825).

5) Line 102- Given that the CD8 T cells that infiltrate the tumor express IFN-gamma, PD-1 is certainly not an exhaustion marker here. PD-1 increases with T cell activation, so in this setting it's simply an activation marker.

We agree and this has been changed in the text (line 164).

6) Line 150 – the Myc / PTEN loss model is interesting, do those cell metastasize in vivo?

We are currently performing a comprehensive characterization of this model in comparison to the wildtype Myc-CaP model. As CP1 is an intra-urethral instillation that ascends to the prostate/tumor, we focused on the effects on the primary orthotopic tumor in this manuscript.

7) In the abstract the authors state that TH17 cells mediate lysis of tumor cells, but this is not documented in the results section as far as I could see. Direct lysis of tumor cells by CD4 T cells is not without precedent, but if the authors wish to claim that their TH17 are mediating lysis additional studies are required. Specifically, they'd need to isolate (sort)

TH17 from tumors (using a bead capture method) and perform an in vitro CTL assay. While that would be an interesting experiment, it's really not required or relevant to the present study. Instead, it might be more facile to simply edit the abstract etc.

In the abstract, Th17 cells are only mentioned when stating “CP1 increased immunogenic cell death of cancer cells, T cell cytotoxicity, and tumor infiltration by activated CD8 and Th17 T cells, mature dendritic cells, M1 macrophages, and NK cells.” We do not have any evidence that Th17 cells directly lyse tumor cells, and we have made a minor text change that we hope now makes it clear that we are only stating that there is increased infiltration of Th17 T cells in CP1-treated tumors (line 12).

Thank you for these reviews and for your consideration. We hope you agree that this manuscript is now suitable for publication in *Nature Communications*.

Reviewer #2 (Remarks to the Author):

In this manuscript, it is demonstrated that the UPEC strain CP1 can accumulate in tumors in the prostate when delivered intra-urethrally. CP1 appears to work synergistically with anti-PD-1 to delay progression by recruiting parts of the immune system. While the numbers of animals are too few in some experiments, this conclusion appears to be likely and is important.

The manuscript has a large number of rectifiable deficiencies. Vital information about the experimental design in many experiments was missing or sufficiently hard to find that I missed it. Claims were unnecessarily exaggerated in places. Trying too hard to impress made some very simple things harder to understand. I persisted because a manuscript that “cut to the chase” would have merit.

A sufficient number of bacteria get to the tumor and persist there to allow a useful immunological reaction. However, what is that number? It was impossible to tell. Near the beginning of colonization, it appears to be 2% of the inoculum can be plated out of the tumor.

We observed an average 3.8×10^6 CP1 in the tumor 9 days after intra-urethral administration (Fig. 1a), or 3.3×10^6 CFU/g tumor (new Fig. 1b) or approximately 2% of the original CP1 inoculation (new Fig. 1c) (9 days after intra-urethral administration is the time point anti-PD-1 antibody treatment is initiated in subsequent experiments). We also performed additional *in vivo* experiments and observed similar values on day 1 after intra-urethral administration (new Supplementary Fig. 3a-c) (lines 97-105).

The 16S qPCR used later needs to be calibrated so can know if there are hundreds or millions of bacteria at later time points.

We performed additional 16S RT-PCR on tumor tissue as well as on titrations of CP1 of known bacteria cell counts ($R^2 = 0.98184$). Tissue 16S RT-PCR values were then calibrated to CP1 cell count, which was subsequently normalized to calibrated PBS-administered tumors, their total RNA yield, as well as the weight of the tumor tissue from which that RNA was extracted. This calibration yielded similar CP1 counts as those determined by tumor tissue culture on both day 1 and day 9 after intra-urethral administration (new Supplementary Fig. 3d) (lines 107-110). All other graphs reporting 16S RT-PCR were normalized to a mouse housekeeping gene, which should

give an accurate relative ratio of the amount of bacteria cells : tumor cells.

What proportion are intracellular?

Quantification of the *E. coli* immunofluorescence identified that approximately 58.2% are extracellular and 41.8% are intracellular (new Fig. 1e quantification) (lines 110-112).

What proportion are still capable of replication? Perhaps there are all persisters or VBNC.

Thank you for this interesting question, as a prior study has suggested the possibility of VBNCs within prostate tumors³. *16S* RNA has been validated as an accurate measure to monitor *E. coli* VBNCs^{4,5}. Therefore, we analyzed CP1 levels by both tumor tissue culture and by *16S* RNA levels within prostate tumors on both day 1 and day 9 after intra-urethral CP1 administration. In addition, we calibrated the *16S* RT-PCR values to CP1 cell counts, as described above.

Overall, CP1 counts derived from *16S* RT-PCR were not significantly different than those attained by tumor tissue culture, signifying that VBNCs are absent or minimal in this model (new Supplementary Fig. 3d). Further, while we cannot control for bacteria lost in the urine or lost from the amplified immune response, the overall CP1 levels did not significantly change over time (new Supplementary Fig. 3d) (lines 107-110). This is consistent with the RT-PCR we performed on later timepoint mice demonstrating that the intra-tumoral bacteria levels (normalized to a mouse housekeeping gene) do not significantly increase over time (Supplementary Fig. 9).

Under many scenarios, it is implausible that ICD is happening on a significant scale, and the pertaining parts could be deleted without any loss of significance of the observation.

We have updated the Results text to emphasize that the ICD reported in Fig. 2a-b is an *in vitro* finding. In this revised manuscript, we now also demonstrated that CP1 induced ICD to a greater degree than MG1655 (new Supplementary Fig. 5b). In addition, to more accurately

represent the CP1 count within the tumor, we again repeated these *in vitro* assays with gentamicin added to the media after the initial 2 hours of co-culture (as done with the gentamicin protection assay. However, it is important to keep in mind that these conditions eliminate the potentially important impact of extracellular CP1 or the ability of CP1 to spread between cells, and therefore may understate the effects of CP1 on inducing ICD). Under these gentamicin conditions, CP1 (but not MG1655) significantly increased calreticulin levels, but did not induce HMGB1 or ATP secretion (new Supplementary Fig. 5c). Finally, and most importantly, we analyzed orthotopic prostate tumor tissue 9 days after intra-urethral CP1 administration and observed a significant increase in HMGB1⁻ nuclei (signifying HMGB1 release, as previously performed⁶) (new Fig. 2c) and areas of increased cell surface calreticulin levels (new Fig. 2d) (lines 124-139).

If the tumors in the CP1 mice were smaller at the time of sacrifice due to sickness, what caused the mice to be sick? Did metastases occur? If the bacteria did not disseminate, metastases might grow freely.

We apologize for any confusion, but analysis of tumor size in both Fig. 4 and Fig. 5 were performed on different mice than those followed for survival analysis. In Figure 4, CP1 was administered on day 8 (after intra-prostatic cancer cell injection), and anti-PD-1 antibody was administered every other day from day 17-29, followed by either sacrificing mice for tumor size analysis on day 30 or the termination of treatment to follow separate mice for survival with no additional intervention. In Figure 5, CP1 was administered on day 9, and anti-PD-1 antibody was administered very other day from day 13-23, followed by either sacrificing mice for tumor size analysis on day 24 or the termination of treatment to follow separate mice for survival. Mice analyzed for tumor size were not sick, and all mice in those graphs received the same treatment timeline and were analyzed the day after treatment termination.

CP1 are localized. This means metastases are not targeted. Is that a good thing or a bad thing?

Systemic CP1 may have resulted in potentially fatal sepsis and would have minimized any future therapeutic potential of administering this microbial agent to patients.

Intra-urethral instillation is one of the strengths of this study, allowing for “local” CP1 delivery into the tumor microenvironment to induce anti-tumor immunity in combination with anti-PD-1 antibody. We are further investigating the potential for the elicited adaptive immune response to target distant lesions, but this is outside the scope of this study that focused on the primary lesion in the orthotopic models.

I would be curious as to how intra-tumor LPS or K12 would behave in this model to dissect out exactly what is necessary for the observations. What happens if you treat with an antibiotic to remove the CP1 after a short time?

Thank you for this suggestion. To answer this, we repeated many of the experiments with CP1 in parallel with MG1655, as it is the prototypical strain from the original patient-derived K-12 *E. coli* that you suggest above. MG1655 has been maintained with “minimal genetic manipulation” and its genome has been sequenced and well categorized¹. *In vitro*, CP1 adhered to, invaded, and intracellularly proliferated in Myc-CaP cells to a greater degree than MG1655 (Supplementary Fig. 2). CP1 also induced ICD to a greater degree than MG1655 (with and without gentamicin) (new Supplementary Fig. 5b-c), CP1 induced caspase 3/7 activity to a greater degree than MG1655 (new Supplementary Fig. 5d-f), and CP1 induced significantly more (but relatively similar) levels of Annexin V early apoptosis in comparison to MG1655 with gentamicin (new Supplementary Fig. 5h). When either CP1 or MG1655 were administered intra-urethrally to orthotopic prostate tumor-bearing mice, only CP1 induced increased levels of both CD8 and CD4 TILs (new Supplementary Fig. 6). With all of these above results comparing CP1 with the K12 *E. coli*, there was no rationale to continue with large scale *in vivo* efficacy studies (lines 124-133, 140-150, 161-162).

While we will continue to explore either LPS or try to identify other potentially immunogenic molecules in this model, LPS, unlike live CP1 bacteria, would not ascend from the urethra to the prostate tumor, and therefore would not reach its target tissue or persist in the mouse. Further, those effects would be limited primarily to TLR4 activation, which would eliminate the many multi-faceted immunomodulatory properties of live CP1 identified in this manuscript (Fig. 8).

Antibiotic treatment is also an interesting question we are pursuing. However, antibiotic treatment would be a major confounding factor, as multiple studies have demonstrated that antibiotic treatment significantly decreases response to PD-1 antibody and immunotherapy^{7,8}, as they alter the gut microbiome that plays a critical role in the systemic efficacy of immune checkpoint inhibitors⁹⁻¹². Therefore, we will be comprehensively analyzing the effects of multiple antibiotics on the efficacy of PD-1 antibody in our model, to then be able to pursue the question of their effect on CP1 clearance, which we believe is outside of the scope of this manuscript. However, we believe that Supplementary Fig. 9b provides evidence that persistent bacterial colonization is necessary for an efficacious synergy with anti-PD-1 antibody, as all long term responders had high bacterial ratios, and all low bacterial ratio mice died early after treatment.

Below I go through the manuscript, line by line.

Lines 64 and 65. Not unique. This is novel but likely not unique. It is not even surprising, in retrospect, given that UPEC invade epithelial cells. It is still important. No need to stretch. Just the facts.

Thank you for the input. We have changed the text and apologize for any over-interpretation anywhere in the manuscript, which we have tried hard to correct.

Line 69. The data regarding virulence factors is based on negative PCR results from other papers. Given the work done on this bacterium, it would be a good idea to spend \$300 on sequencing the genome.

We performed whole genome sequencing of CP1 (Supplementary Fig. 1a) and used the sequencing to do phylogenetic group analysis, MLST analysis, virulence factor analysis, and genome comparative analysis to MG1655 (Supplementary Fig. 1b-c) (lines 67-78, 354-377).

Line 70: It is claimed that the specificity for prostate cancer cells is being tested. This is an exaggeration. No evidence is provided that CP1 is any different from any other UPEC in its abilities. Indeed, its genetics may mean it is “worse” than many other UPEC, which could be a good thing for a living therapeutic.

We removed specificity from the text. We believe that CP1 is a UPEC that is representative of a class of immunogenic and tissue-specific bacteria that can potentially be similarly exploited for this purpose, and we are not claiming it is the only UPEC that can strongly adhere/invade/proliferate in prostate cancer cells. We apologize for the undue emphasis on specificity.

Line 73: Fig 1a. The laboratory K12 strain does not adhere well, does not invade, and does not proliferate intracellularly. UPEC do. This is a straw man. Supplement.

This has been moved to Supplementary Fig. 2.

Line 74: It is not clear if Fig. 1b was done in Gentamycin or not, and for how long. I can comment when I know. Supplement.

This (now Supplementary Fig. 5a) was not performed in gentamicin. We hope it is now clearer which *in vitro* assays were (Supplementary Fig. 2, Supplementary Fig. 5c,e,f,h,i) or were not (Fig. 2a-b, Supplementary Fig. 5a,b,d,g) performed in gentamicin (added 2 hours after initiation of the co-culture).

Line 79. The timing and units really matter because overall, it is not clear how much proliferation is taking place, if any. I am guessing this was about two days after administration, at which point bacteria representing 2% of the inoculum were found in the tumor.

CFU values from Fig. 1a were analyzed from tumors 9 days after intra-urethral CP1 administration, not 2 days. Further, we performed additional culture analysis and 16S RT-PCR analysis of tumors on both day 1 and day 9 after intra-urethral CP1 administration, and observed similar bacteria values at each timepoint, both representing approximately 2% of the initial inoculum (new Supplementary Fig. 3).

In summary for this part of Figure 1:

“UPEC colonize the urinary tract and are known to invade and proliferate in epithelial cells (REFS). CP1 was isolated from a prostate and is in a rare UPEC phylogenetic group that lacks many of the virulence factors found in other phylogenetic groups (REF), so it is potentially less virulent. We demonstrated that the ability to invade epithelial cells extends to a mouse epithelial prostate cancer cell line, at least in the case of CP1. (supp figure 1a and b). It has been established that an intraurethral dose of $\approx 10^8$ UPEC bacteria, including CP1 (REF), leads to colonization of the bladder and prostate without progressing to the kidney or progressing to the systemic tissues. Using this delivery

model, we reiterated these results and additionally showed that CP1 can be isolated from an orthotopically implanted syngeneic prostate tumor. Delivery was X days after implantation and recovery was Y days later and over 10^6 bacteria per (what units?) were recovered (grams? Tissue, what?) (Figure 1c, now a).”

The whole of the results need the same streamlining and clarity.

Thank you for the suggestion, we have made changes throughout the results section to simplify and clarify the figures in line with your above suggestion, including adding additional details about methodology, treatment timelines, and bacteria values, and hope that the results are now clearer to understand.

Line 80. There is no hint as to why Fig. 1d shifts to a 16S PCR assay from CFUs. Explain why. Here and elsewhere, there is a need to calibrate the number of bacteria with the reference gene and with the number of host cells in the sample. We will learn later that tumors vary a lot in size, partly depending on the treatment. Without knowing the size of the sample (and a calibration to approximate CFU) we have no idea how many bacteria are present and whether variation is simply because non-replicating bacteria are diluted to different degrees.

16S was performed as an additional validating readout of tracking successful colonization of CP1 in the tumor after intra-urethral administration. All graphs reporting *16S* RT-PCR data were already normalized to a mouse housekeeping gene, giving a ratio that should be representative of the number of host cells in the sample. Further, we performed additional *16S* RT-PCR on known cell count titrations of CP1 to create a ladder ($R^2 = 0.98184$) with which we calibrated new tissue *16S* RT-PCR values to CP1 cell counts, and then normalized those counts to calibrated PBS-administered tumors, total RNA yield, and the weight of the tissue from which that RNA was extracted. This experiment demonstrated that CFUs calibrated from *16S* RT-PCR were similar to those derived by tumor tissue culture (new Supplementary Fig. 3d) (lines 107-110).

Line 81. In fig. 1e, we see some green and yellow spots. What is the proportion inside and out? Is there an estimate of how many bacteria are present? How long after the tumor was formed did the delivery of bacteria take place? How long after delivery was the imaging was performed?

Is there any evidence that the bacteria divide after they arrive at the prostate?

As stated above, CP1 was administered 8 days after cancer cell injection (to first allow for tumor

establishment). CFU and *E. coli* imaging analyses in Fig. 1 were performed 9 days after CP1 administration (day 17 total after initial intra-prostatic tumor cell injection, the same day as Fig. 3 tumor immune analysis and the same day when anti-PD-1 antibody administration begins in Fig. 4). We identified an average 3.8×10^6 total CP1 CFUs (Fig. 1a), or 3.3×10^6 CFU/g tumor (new Fig. 1b). Quantification of the *E. coli* immunofluorescence identified that approximately 58.2% are extracellular and 41.8% are intracellular (new Fig. 1e quantification) (lines 97-102, 110-112).

An additional analysis comparing day 1 after intra-urethral CP1 administration (day 9 total after intra-prostatic cancer cell injection) with day 9 after CP1 administration (day 17 total after intra-prostatic cancer cell injection) demonstrated no significant change in intra-tumoral CP1 levels over that time (new Supplementary Fig. 3a-c), and calibrating *I6S* RNA values at each timepoint resulted in a similar CP1 counts as those attained by tissue culture (new Supplementary Fig. 3d) (lines 102-110).

Line 90. It is hard to interpret figure 2 reliably without knowing whether this involved a gentamicin protocol or not. What were the number of bacteria inside and outside the host cell at 72hrs? If there was no gentamicin then one has to imagine that 10^{10} cfu/ml of a control K12 bacterium after 72hrs might have a similar adverse effect. Fitting this experiment into the subsequent theories is dependent on how many bacteria are present at 72 hours in this experiment and how many are present within the tumor and replicating in the in vivo experiment.

Fig. 2 and new Supplementary Fig. 5b did not include gentamicin, and both CP1 and MG1655 were about $1.5-2 \times 10^8$ CFU/ml at 72 hours, which we agree is not physiologically relevant to what is occurring in the tumor. To answer your above questions, we repeated the assays with both CP1 and MG1655, both with and without gentamicin, at an initial MOI of 1 (50,000 Myc-CaP cells and 50,000 bacteria). With gentamicin after 72 hours, there was an average 1,056,000 Myc-CaP cells per well and an average 5,440 CP1 cells per well (10.9% of the original 50,000), resulting in a final CP1:Myc-CaP ratio of 0.005. This should be more representative of the *in vivo* tumor experiments, and, as we state in the text, may also understate the effects of CP1, because in addition to decreasing total CP1 count, this gentamicin condition also

eliminated any potential importance of extracellular CP1 interacting with tumor cells or CP1 spreading between cells.

Without gentamicin, CP1 induced ICD (HMGB1, ATP, and calreticulin) to a greater degree than did MG1655 (new Supplementary Fig. 5b). With gentamicin, CP1 (but not MG1655), still increased calreticulin⁺ cancer cells, but did not induce HMGB1 or ATP secretion (new Supplementary Fig. 5c). We also emphasize in the text that these are *in vitro* findings. To further support the *in vitro* findings, we then analyzed tumor tissue 9 days after intra-urethral CP1 administration and observed a significant increase in HMGB1⁺ nuclei (signifying HMGB1 release, as previously performed⁶) (new Fig. 2c) and areas of increased cell surface calreticulin levels (new Fig. 2d). And as you mention above, ICD is only one of the potential mechanism of actions of CP1, and is not critical to the overall findings of this manuscript (lines 124-139).

Line 101. Fig. 3. It is not clear to me how long after tumors were introduced the bacteria were delivered, and how long after that delivery these immunophenotyping measurements were performed. How was this time chosen? Surely, it matters. Presumably, there is a time component to this response. Was this explored?

CP1 was administered on day 8 after intra-prostatic cancer cell injection, to allow adequate time for tumor establishment. This timepoint was chosen with a timecourse of *in vivo* IVIS imaging and preliminary experiments to verify tumor establishment by this time. Immunophenotyping was then performed 9 days after intra-urethral CP1 administration (17 days total after intra-prostatic cancer cell injection), allowing adequate time for CP1 to induce any potential tumor infiltrating adaptive immune response before either immune analysis or beginning anti-PD-1 antibody administration. This timeline was also created with careful regard to the time constraints and lifespan when working with this aggressive tumor model.

Line 125. It would be good to know the actual p value for the the KM plot.

$P = 0.0066$ comparing Unt to CP1 + PD-1 in Fig. 4a, which has been added to the Results text.

Line 129. I presume these experiments are done at the time of sickness-induced sacrifice.

What happened to mice that never became sick during the experiment?
IVIS was done before treatment. It was then done again but it is not clear when that was.
Presumably not at the time of death when the tumor can be measured directly?

We apologize for the confusion, but as stated above, analysis of tumor size in both Fig. 4 and Fig. 5 were performed on different mice from those followed for survival analysis. For tumor size analysis, all mice were euthanized and tumors analyzed on the day after the final anti-PD-1 antibody administration, and they were not at the stage of sickness-induced sacrifice. Mice analyzed for survival were different mice treated with the same timeline of CP1 and anti-PD-1 antibody, but were then followed without any additional intervention until sickness-induced sacrifice. Overall, there were only 1 or 2 mice/group that never reached sickness-induced sacrifice in the survival analysis.

IVIS was performed before treatment and then again after the termination of treatment. This was performed near the time of sacrifice (for mice analyzed for tumor size), because, unlike with direct tumor measurements, this IVIS analysis allowed for normalization to pre-treatment tumor imaging to normalize for any variation in pre-treatment tumor burden. IVIS analysis also differs from direct tumor measurements in that it quantifies the level of live tumor cells, while direct tumor measurements do not account for any inner necrotic or dead tissue. All of these analyses serve to complement and validate each other and reinforce the finding of the efficacy of CP1 with anti-PD-1 antibody.

Line 132. Supplementary fig. 4. These are ratios, not levels. The tumors could be of various sizes and perhaps smaller in animals that survive longer. To be careful, it would be interesting to normalize for sample volume or sample weight and plot again for the dose of CP1 present per mouse at the time of sacrifice.

We agree and have changed the text and removed levels. In this survival analysis, overall tumor size was similar between groups at the time of sacrifice. However, the tissue taken for RNA extraction was not weighed. Therefore, all *16S* RT-PCR were normalized to a mouse housekeeping gene, which should give an accurate ratio of bacteria : tumor cells, comparable to normalizing by tissue size or weight. In the analysis in Fig. 1b and Supplementary Fig. 3b, we normalized tissue culture CFU by tumor tissue

weight, and in Supplementary Fig. 3d, we calibrated *16S* RT-PCR values to CP1 bacteria counts normalized to tumor weight in comparison to the culture counts.

It is a pity there is no experiment with another bacteria, such as Nissle, or different doses, or an antibiotic to clear bacteria earlier, or an IV delivery, or LPS delivered intra-tumor, to try to demonstrate that this particular route of delivery or this particular bacterium is of particular merit.

As stated above, per your suggestion, we repeated many of the experiments with CP1 in parallel with the K-12 MG1655, the prototypical strain from the original patient-derived K-12 *E. coli* that has been maintained with “minimal genetic manipulation” and whose genome has been sequenced and well categorized¹. *In vitro*, CP1 adhered to, invaded, and intracellularly proliferated in Myc-CaP cells to a greater degree than MG1655 (Supplementary Fig. 2). CP1 also induced ICD to a greater degree than MG1655 (with and without gentamicin) (new Supplementary Fig. 5b-c), CP1 induced caspase 3/7 activity to a greater degree than MG1655 (new Supplementary Fig. 5d-f), and CP1 induced significantly more (but relatively similar) levels of Annexin V early apoptosis in comparison to MG1655 with gentamicin (new Supplementary Fig. 5h). When either CP1 or MG1655 were administered intra-urethrally to orthotopic prostate tumor-bearing mice, only CP1 induced increased levels of both CD8 and CD4 TILs (new Supplementary Fig. 6). With all of these above results comparing CP1 with the K12 strain, there was no rationale to continue with large scale *in vivo* efficacy studies (lines 124-133, 140-150, 161-162).

As we stated in a previous reply, antibiotic treatment would be a major confounding factor, as multiple studies have demonstrated that antibiotic treatment significantly decreases response to PD-1 antibody and immunotherapy^{7,8} and have highlighted the critical role an intact gut microbiome plays in the systemic efficacy of immune checkpoint inhibitors⁹⁻¹². Therefore, we will be comprehensively analyzing the effects of multiple antibiotics on the efficacy of PD-1 antibody in our model, to then be able to pursue the question of their effect on CP1 clearance, which we believe is outside of the scope of this manuscript. However, we believe that Supplementary Fig. 9b provides evidence that persistent bacterial colonization is necessary for an efficacious synergy with anti-PD-1

antibody, as all long term responders had high bacterial load, and all low bacterial load mice died early after treatment.

I.V. delivery would not be possible with this *E. coli* model to avoid sepsis. The intra-urethral delivery method is one of the strengths of this paper, and has strong translatable clinical potential when considering CP1 as a therapeutic for patients, as intra-urethral instillation allows for CP1 colonization of the prostate tumor and its induction of an anti-tumor immune response without causing sepsis or systemic toxicities.

Also as stated above, while we will continue to explore either LPS or try to identify other potentially immunogenic molecules in this model, LPS, unlike live CP1 bacteria, would not ascend from the urethra to the prostate tumor, and therefore would not reach its target tissue or persist in the mouse. Intra-tumoral injection would involve a second major survival surgery or would involve trans-rectal or trans-perineal imaging guided injection beyond the scope of what is available for pre-clinical murine models. Further, those effect would be limited predominantly to TLR4 activation, which would eliminate the many multi-faceted immunomodulatory properties of live CP1 identified in this manuscript (Fig. 8).

Line 154. When the word “significantly” is used, then a P value is desirable, especially as only the range is given in the figures. This experiment has marginal p values and there are only three samples in each treatment group in 5f, when seven were theoretically available. What happened to the one mouse in the critical CP1+PD-1 group in the KM plot?

P values have been added throughout the Results text when significantly is used. Also, as stated above, mice followed for survival analyses were different mice than those euthanized the day after the final day of treatment and analyzed for tumor volume. In the survival analyses, rarely, mice such as the one mouse in the CP1+PD-1 group in the KM plot was censored, either due to death from mice fighting or if analysis revealed death due to non-orthotopic tumor seeding on the anterior abdominal wall blocking urine outflow despite a small primary orthotopic prostate tumor.

Line 222. There is no direct evidence that there are enough actively dividing bacteria in

the tumor to cause this. The *in vitro* experiments may contain many orders of magnitude more bacteria per host cell than the tumors, until it is demonstrated otherwise. The data may exist but I cannot see it.

We have now emphasized in the text that *in vitro* ICD findings are *in vitro*. However, we also now demonstrate in new Supplementary Fig. 5c that greatly decreased levels of CP1 (cultured with gentamicin) were still able to increase calreticulin⁺ cells (after 72 hours the CP1:Myc-CaP ratio was 0.005, with the surviving intracellular CP1 representing approximately 10.9% of the original 50,000 cells added per well, which was multiple orders of magnitude less bacteria than in Fig. 2a-b at 72 hours). Most importantly, in new Fig. 2c-d we demonstrate that CP1 can induce ICD *in vivo* in tumors (increased HMGB1⁻ nuclei and areas of increased cell surface calreticulin levels) (lines 126-139).

Line 224. There is no necessity for these bacteria to be delivered IV. It is done to seek out metastases. Maybe someone should try to delivery these other bacteria intra-urethrally. Indeed, the fact that CP1 does not appear to seek out metastases could be construed as a weakness.

The text has been changed accordingly.

Line 225. Surely genetic manipulations in Enterobacteria are straightforward, and you later propose to do precisely that.

Yes, future work will focusing on genetic manipulations to increase immunostimulatory efficacy, bacterial tracking, and bacterial attenuation, but that is outside the scope of this manuscript.

Line 225. Not consistently (there is overlap) but significantly.

We believe you are referring to line 241 (now line 308: “a single dose of CP1 consistently augmented the anti-tumor response to...”), and that line has been changed accordingly.

Line 285. The papers say 10^8 and here it seems to be 2×10^8 . Clarify.

2×10^8 CP1 in 10 μ l are injected in each intra-urethral administration.

Line 286. It costs less than \$300 to sequence the genome.

We performed whole genome sequencing of CP1 (Supplementary Fig. 1a) and used the sequencing to do phylogenetic group analysis, MLST analysis, virulence factor analysis, and genome comparative analysis to MG1655 (Supplementary Fig. 1b-c). We discovered that CP1 is actually in the B2, not B1 phylogenetic group. Recent publications have demonstrated that the Clermont PCR primers resulted in 14.0% misclassifications, with failure to amplify *chuA* and *yjaA*¹³, which would result in the misclassification of a B2 *E. coli* as B1. Thank you for the suggestion to sequence the genome (lines 67-78, 354-377).

Line 288. How many tumor cells?

1×10^6 Myc-CaP cells were plates in this experiment with CP1 or MG1655 at an MOI of 1.

Line 298: Typo. CPI should read CP1.

Thank you for your thorough review, this error has been corrected.

Line 769. What are the number of living replicating bacterial cells at any time in the tumor? Is this number sufficient to demonstrate that the bacteria cause cell death in the tumor, or is it merely the presence of low levels of an extremely immunogenic bacterium that is the trigger?

We observed an average 3.8×10^6 total CP1 colony forming units (CFUs) (Fig. 1a), or 3.3×10^6 CFU/g tumor (Fig. 1b) from the tumor 9 days after intra-urethral CP1 administration. To answer your questions, we now demonstrate that decreased levels of CP1 (cultured with gentamicin, after 72hrs CP1:Myc-CaP ratio of 0.005, with the surviving intracellular CP1 representing approximately 10.9% of the original 50,000 cells added per well, multiple orders of magnitude less bacteria than in Figure 2a-b and new Supplementary Fig. 5b) were still able to increase calreticulin surface expression (new Supplementary Fig. 5c). Importantly, intra-urethral CP1 also induced ICD in tumors (increased the percentage of HMGB1⁻ nuclei and areas of increased cell surface calreticulin levels) (new Fig.

2c-d). We believe that these effects are a combination of CP1's ability as a UPEC to colonize the prostatic tissue as well as its high immunogenicity, and that these are not mutually exclusive characteristics, but rather a pair of traits that should be sought after to identify similar future microbes (lines 97-102, 126-139).

Thank you for these reviews and for your consideration. We hope you agree that this manuscript is now suitable for publication in *Nature Communications*.

Reviewer #3 (Remarks to the Author):

Using two orthotopic prostatic adenocarcinomas (Myc driven or PTEN deficient and Myc driven) inoculated by intra-urethral administration, the authors demonstrate that i) the combination of a live B1 phylogenetic group uropathogenic E. coli (UPEC) called « CP1 » synergizes with anti-PD1 Ab to reduce tumor growth and increase survival, ii) to reprogram the TME (influx of TH17 and Tc1 TIL, decrease local IL-6/VEGF, increase CXCL10, immunogenic cell death), iii) by enhancing the recruitment from draining LN (inhibition of lymphocyte egress from dLN using FTY720 blunts the TIL accumulation and survival benefit).

This is an interesting study that deserves a particular attention. However, additional controls are missing from this first version that need to be brought up to make it more attractive.

Major comments :

1/ Potential experimental flaws : 1A/For all figures where TIL phenotypes are shown, some assays are depicted in %, others are in cells/mm³, while tumor sizes are not the same inbetween groups (timing of Facs at sacrifice).One needs to represent all cell types within tumors of quite similar volume, at early stages of treatment start, to avoid any flaws related to tumor volume (of course, T cells are sparse in a huge tumor, even when this one has responded earlier one). Fig. 7 may be subjected to this specific criticism.

To most accurately represent the flow cytometry data, we performed the flow cytometry after measuring live cell counts within each sample, thereby allowing us to generate data on infiltrating cell densities as accurate as possible (cells/mm³), as has been similarly published¹⁴⁻²⁰. To additionally then analyze the immune phenotype of specific infiltrating immune cell subtypes, we also represented some graphs as percentages of their parent gate. We believe that both forms of data add value, and we hoped to make this clear by plotting cell densities (scatter plots) differently from percentages of parent gates (scatter box plots with adjacent representative flow cytometry plots). We believe that both metrics are accurate and importantly analyze separate aspects of the tumor immune infiltrate.

With regards to the tumor size at the time of flow cytometry analysis, the majority of the flow cytometry data reported in this manuscript (Fig. 3), was performed at early stages of treatment start, as you suggest (9 days after intra-urethral CP1 administration, the day that anti-PD-1

antibody administration begins in Fig. 4). At this early stage, there is no significant difference in tumor size between PBS vs CP1 tumors.

For Fig. 7, we had already demonstrated that CP1 increases T cells in PTEN KO tumors, and we now wanted to determine the necessity of this T cell recruitment for its efficacy with PD-1 blockade. The flow cytometry was performed mainly as a control to demonstrate that CP1 again increased TILs as expected and that the FTY720 drug successfully inhibited T cell egress from the lymph nodes into the tumor microenvironment. We believe that the summation of this manuscript adequately demonstrates that CP1 increases TILs, and it is not that T cells are sparse in untreated tumors because they are bigger, but rather that untreated tumors are bigger because of their low baseline level of TILs.

1B/ in most fig. legends, the authors failed to announce how many experiments were run, how many mice/group, while some graphs or bars depict few dots...(except Fig 4).

We believe that most figure legends do include these details (bold in Figure 1 legend below, for example), but we have also added more details throughout the legends wherever applicable, which we hope are now satisfactory. We report technical repeat numbers, biological repeat numbers, and independent experiment repeat numbers throughout the figures, as well as mice/group numbers for all data derived from *in vivo* experiments.

Figure 1: (a) Gentamicin protection assay with CP1 and MG1655 with Myc-CaP cells *in vitro*, reported as colony forming units (CFUs), **performed in sextuplicates, plated in serial dilutions.** (b) LDH level, as a measure of cell death, from CP1 and Myc-CaP co-culture, **performed in triplicates.** (c) Bacterial colonization in the bladder, prostate tumor, ipsilateral and contralateral kidneys, liver, and spleen, **performed in biological triplicates, plated in serial dilutions.** (d) *16S* qRT-PCR of tumor RNA, normalized to *RPLP0*, **performed in biological quadruplicates, technical duplicates.** (e) *E. coli* IF of tumor tissue (green = intracellular, yellow = extracellular), (scale bar, 20 μ m; magnified scale bar, 4 μ m). **Mice $n = 4-5$, performed in 2 independent experiments.** Data represented as mean \pm S.E.M. Statistical significance was

determined by Student's *t*-test. * $P < 0.05$, ** $P < 0.01$, *** $P < 0.001$, **** $P < 0.0001$.

2/ Since many prostatic cancers result from chronic prostatitis, why is the tumor microenvironment so « cold » in most of them? does it mean that the spectrum of bacteria naturally invading prostate tumors is immunosuppressive ? or that prostate tumors are not normally invaded by bacteria ? or microbes. Authors should cultivate a couple of human primary prostate cancers arising from chronic inflammation or at least performed a FISH analysis of prostate cancer tissues to visualize bacterial products in situ, as already reported for pancreatic cancers (Geller, L. T. et al. Science 357, 1156–1160 (2017)).

That is a very interesting question, because as with most tissue types, there is a strong association between chronic inflammation (possibly due to microbial infection, but also possibly caused by uric acid crystals from urine reflux, estrogen, diet (heterocyclic amines), obesity, physical trauma (corpora amylacea), or treatments²¹) and tumorigenesis. However, it is very important to consider the context of the inflammation, and differentiate between chronic inflammation speeding up tumor development in a potentially genetically pre-disposed tissue versus acute inflammation or immune checkpoint inhibitors driving or amplifying an adaptive immune response. In our model, CP1 is a new bacteria administered to an already established tumor to overcome the existing immunosuppression. Any previous bacteria in the tumor may have potentially been involved in chronic inflammation driving tumorigenesis, but are not driving any strong immune response in the established tumor.

It has recently become evident that many tissue and cancer types contain previously underappreciated microbiomes. As a result, prior studies have performed your above suggested experiment to analyze if prostate tumors contain any microbial flora. An initial PCR and culture analysis of 170 core samples from 30 prostate cancer patients identified bacterial DNA in 87% of tumors and a total of 83 distinct microorganisms. However, no individual bacterial species was associated with acute or chronic prostatic inflammation³. A more recent and intricate ultradeep pyrosequencing analysis of 16 prostatectomy specimens determined that *Propionibacterium* spp. was the most numerous bacterial genera within prostate tumors, *Staphylococcus* spp. was found more frequently within the tumor and peri-tumoral region than within non-tumor tissue, and *Streptococcus*

spp. was more frequently found in non-tumor tissue²².

While it is unclear if any of these bacteria were also involved in driving tumorigenesis, to date no single microbial driver has been identified linking chronic prostatitis with tumor formation²³. However, the presence of these bacteria in the tumor microenvironment does not appear to increase the anti-tumor immune response, as numerous prior studies have comprehensively and consistently demonstrated the scarcity of TILs in the prostate tumor microenvironment. Still, these bacteria might be functionally pro-inflammatory, and may be responsible for the high level of infiltration of immunosuppressive MDSCs, Tregs, and M2 macrophages reported in prostate tumors.

3/ Specificity : it is intriguing to observe that CP1 is effective at i) homing only at prostate cancer tissues and not kidney, ii) at promoting ICD, iii) ideally reprogramming the TME, iv) synergizing with aPD1 Ab, with no toxicity. Given this outstanding constellation, one would like to appreciate that any random bacterium would not perform as well and that these properties are somehow quite « unique ».

A « negative Gram negative control bacterium» as well as other *E. coli* clones or isolates should be utilized for all experiments or at least in the Myc driven tumor model.

Thank you for this suggestion, we agree that this is an important control to include for this study. Therefore, we repeated many of the experiments with CP1 in parallel with MG1655, as it is the prototypical strain from the original patient-derived K-12 *E. coli* that has been maintained with “minimal genetic manipulation” and whose genome has been sequenced and well categorized¹. *In vitro*, CP1 adhered to, invaded, and intracellular proliferated in Myc-CaP cells to a greater degree than MG1655 (Supplementary Fig. 2). CP1 also induced ICD to a greater degree than MG1655 (with and without gentamicin in the media to limit bacteria count) (new Supplementary Fig. 5b-c), CP1 induced caspase 3/7 activity to a greater degree than MG1655 (new Supplementary Fig. 5d-f), and CP1 induced significantly more (but relatively similar) levels of Annexin V early apoptosis in comparison to MG1655 with gentamicin (new Supplementary Fig. 5h). When either CP1 or MG1655 were administered intra-urethrally to orthotopic prostate tumor-bearing mice, only CP1 induced increased levels of both CD8 and CD4 TILs (new Supplementary Fig. 6). With all of these above results comparing CP1 with the K12 strain, there was no rationale

to continue with large scale *in vivo* efficacy studies (lines 124-133, 140-150, 161-162).

It is also important to keep in mind that while we think CP1 is a unique microbe, we also believe that it is representative of a class of patient-derived bacteria that may also possess strong tissue-tropic and immunomodulatory properties and can similarly be exploited as cancer immunotherapeutics in future studies.

4/ Dose effects : it would be elegant to show a dose-effect of CP1 together with a constant dosing of anti-PD1 Ab.

Past intra-urethral uropathogenic bacterial *in vivo* studies have demonstrated that bacterial instillation leads to murine tissues colonized with distinct either low or high CFU values, but rarely intermediate CFU values. With regard to your comment, increased the inoculum dose of bacteria administered did not increase either the percentage of mice colonized or the high CFU values in those that were colonized. Superinfection with multiple repeat bacterial inoculations at various timepoints also did not increase the CFU values of highly colonized mice relative to tissues after single inoculation. Overall, these data indicate that infected tissues reached maximum colonization levels from the initial inoculum, which did not increase with higher doses²⁴.

Similarly, we did not observe any increase in the level of high bacterial burden in CP1-colonized tumors over time, also indicating a maximum tissue colonization had been reached (Supplementary Fig. 9b). Further, we performed an additional *in vivo* experiment comparing CP1 bacterial load in the orthotopic prostate tumor on day 1 versus day 9 after intra-urethral administration, and determined that CP1 colonization levels did not significantly change over time (new Supplementary Fig. 3) (lines 102-105). Overall, these prior studies and our data provide strong indication that intra-urethral UPEC colonization occurs in an all-or-nothing manner, and colonized tissues that reach high bacterial burden and CFUs are already at a maximal level, which would not be increased with higher doses of initial CP1 administration.

5/ Th2 microenvironments : from the cytokine arrays, one can see IL-5 and IL-9 dominating the scenario. This atmosphere could favor the differentiation of authentic

TH9 cells. Can IL-21 blockade antagonize the synergistic effects ? is PU-1 (pathognomic of TH9 cells) overexpress in those prostatic TILs ?

IL-9 is only significantly upregulated in Fig. 2e. However, that figure represents *in vitro* Myc-CaP cancer cell-derived cytokines after CP1 exposure (no immune cells were present in this experiment). While IL-9 is the predominant cytokine secreted from Th9 T cells, it is not involved in inducing their differentiation. In all subsequent cytokine arrays in this manuscript that did involve tumor tissue with any infiltrating T cells, IL-9 levels were not increased (Fig. 3n, Supplementary Fig. 10a-b). Also, these arrays did not identify any increase in IL-10, which can also be produced by Th9 T cells²⁵. Overall, there is no evidence of increased Th9 cells after CP1 administration.

IL-5 is expressed from Th2 T cells, but is not involved in inducing Th9 T cell differentiation, and is also not secreted by Th9 cells, as PU.1 inhibits GATA-3 activity²⁶, which is required for IL-5 expression^{27,28}. In addition, while IL-5 is the most upregulated cytokine in Fig. 3n, it is relatively unchanged in Fig. 2c, Supplementary Fig. 10a-b.

On the contrary to IL-9 and IL-5, IL-4 is a critical cytokine for the induction of Th9 cells^{25,29,30}. However, this cytokine is not significantly upregulated in any of the cytokine assays in this manuscript.

Finally, we performed *PU.1* RT-PCR from CP1-treated orthotopic Myc-CaP tumor tissue, and observed no difference in the level of *PU.1* expression between the two groups (graph below). Therefore, we don't believe that a therapeutic *in vivo* orthotopic tumor experiment with IL-21 blockade is warranted.

Minor comment:

It is surprising that *E. coli*, which is not expected to be an intracellular bacterium, is

cytopathogenic (LDH release). Can be authors use a battery of assays that would be convincing that « typical or atypical » cell death is occurring, such as caspase 3 or 8 cleavage, annexinV/PI stainings, PARP cleavage, RIPK1-3 activation, MLKL phosphorylation...

In this manuscript we identify that immunogenic cell death is strongly induced in tumor cells after exposure to CP1. In this revision we also now include data demonstrating that CP1 can induce ICD to a greater degree than the control MG1655 *E. coli* (new Supplementary Fig. 5b), CP1 can still induce ICD in media with gentamicin (leaving greatly decreased and only intracellular bacteria) (new Supplementary Fig. 5c), as well as evidence for CP1 inducing ICD in *in vivo* tumor tissue, with an increase in HMGB1⁻ nuclei (signifying HMGB1 release, as previously performed⁶) (new Fig. 2c) and areas of increased cell surface calreticulin levels (new Fig. 2d) (lines 124-139).

We also performed a battery of cell death assays, as per your suggestion. CP1, but not MG1655, induced caspase 3/7 activity (new Supplementary Fig. 5d, normalized to cell count via MTT assay to control for loss of Myc-CaP cells) without gentamicin. With gentamicin, only CP1 induced caspase 3/7 activity after 6 hours (new Supplementary Fig. 5e), while both CP1 and MG1655 induced caspase activity at 24 hours (new Supplementary Fig. 5f). In addition, both bacteria induced a late apoptosis phenotype (Annexin V⁺ PI⁺) without gentamicin (Supplementary Fig. 5g) and an increased early apoptosis phenotype (Annexin V⁺ PI) with gentamicin (new Supplementary Fig. 5h, CP1 early apoptosis significantly greater than MG1655). Finally, neither CP1 nor MG1655 with gentamicin increased MLKL phosphorylation, RIP1 levels, or PARP cleavage from Myc-CaP cells (new Supplementary Fig. 5i. Bacterial co-cultures without gentamicin interfered with protein loading for western blots). Overall, CP1 induced caspase activity, and its ability to induce ICD appears to be occurring in a necroptosis-independent manner⁶ (lines 140-150).

Thank you for these reviews and for your consideration. We hope you agree that this manuscript is now suitable for publication in *Nature Communications*.

References

- 1 Blattner, F. R. *et al.* The complete genome sequence of Escherichia coli K-12. *Science* **277**, 1453-1462 (1997).
- 2 Rudick, C. N. *et al.* Uropathogenic Escherichia coli induces chronic pelvic pain. *Infection and immunity* **79**, 628-635, doi:10.1128/IAI.00910-10 (2011).
- 3 Sfanos, K. S. *et al.* A molecular analysis of prokaryotic and viral DNA sequences in prostate tissue from patients with prostate cancer indicates the presence of multiple and diverse microorganisms. *The Prostate* **68**, 306-320, doi:10.1002/pros.20680 (2008).
- 4 Yaron, S. & Matthews, K. R. A reverse transcriptase-polymerase chain reaction assay for detection of viable Escherichia coli O157:H7: investigation of specific target genes. *J Appl Microbiol* **92**, 633-640 (2002).
- 5 Ramamurthy, T., Ghosh, A., Pazhani, G. P. & Shinoda, S. Current Perspectives on Viable but Non-Culturable (VBNC) Pathogenic Bacteria. *Front Public Health* **2**, 103, doi:10.3389/fpubh.2014.00103 (2014).
- 6 Yang, H. *et al.* Contribution of RIP3 and MLKL to immunogenic cell death signaling in cancer chemotherapy. *Oncoimmunology* **5**, e1149673, doi:10.1080/2162402X.2016.1149673 (2016).
- 7 Routy, B. *et al.* Gut microbiome influences efficacy of PD-1-based immunotherapy against epithelial tumors. *Science* **359**, 91-97, doi:10.1126/science.aan3706 (2018).
- 8 Thompson, J., Szabo, A., Arce-Lara, C. & Menon, S. P1.07-008 Microbiome & Immunotherapy: Antibiotic Use Is Associated with Inferior Survival for Lung Cancer Patients Receiving PD-1 Inhibitors. *Journal of Thoracic Oncology* **12**, S1998, doi:10.1016/j.jtho.2017.09.926.
- 9 Gopalakrishnan, V. *et al.* Gut microbiome modulates response to anti-PD-1 immunotherapy in melanoma patients. *Science* **359**, 97-103, doi:10.1126/science.aan4236 (2018).
- 10 Vetizou, M. *et al.* Anticancer immunotherapy by CTLA-4 blockade relies on the gut microbiota. *Science* **350**, 1079-1084, doi:10.1126/science.aad1329 (2015).
- 11 Sivan, A. *et al.* Commensal Bifidobacterium promotes antitumor immunity and facilitates anti-PD-L1 efficacy. *Science* **350**, 1084-1089, doi:10.1126/science.aac4255 (2015).
- 12 Routy, B. *et al.* Gut microbiome influences efficacy of PD-1-based immunotherapy against epithelial tumors. *Science*, doi:10.1126/science.aan3706 (2017).
- 13 Turrientes, M. C. *et al.* Recombination blurs phylogenetic groups routine assignment in Escherichia coli: setting the record straight. *PloS one* **9**, e105395, doi:10.1371/journal.pone.0105395 (2014).
- 14 Manuel, E. R. *et al.* Enhancement of cancer vaccine therapy by systemic delivery of a tumor-targeting Salmonella-based STAT3 shRNA suppresses the growth of established melanoma tumors. *Cancer research* **71**, 4183-4191, doi:10.1158/0008-5472.CAN-10-4676 (2011).

- 15 Galarneau, H., Villeneuve, J., Gowing, G., Julien, J. P. & Vallieres, L. Increased glioma growth in mice depleted of macrophages. *Cancer research* **67**, 8874-8881, doi:10.1158/0008-5472.CAN-07-0177 (2007).
- 16 Huang, Y. *et al.* CD4+ and CD8+ T cells have opposing roles in breast cancer progression and outcome. *Oncotarget* **6**, 17462-17478, doi:10.18632/oncotarget.3958 (2015).
- 17 Kawakubo, M., Cunningham, T. J., Demehri, S. & Manstein, D. Fractional Laser Releases Tumor-Associated Antigens in Poorly Immunogenic Tumor and Induces Systemic Immunity. *Scientific reports* **7**, 12751, doi:10.1038/s41598-017-13095-8 (2017).
- 18 Ehrig, K. *et al.* Growth inhibition of different human colorectal cancer xenografts after a single intravenous injection of oncolytic vaccinia virus GLV-1h68. *J Transl Med* **11**, 79, doi:10.1186/1479-5876-11-79 (2013).
- 19 Gonzalez-Carmona, M. A. *et al.* CD40ligand-expressing dendritic cells induce regression of hepatocellular carcinoma by activating innate and acquired immunity in vivo. *Hepatology* **48**, 157-168, doi:10.1002/hep.22296 (2008).
- 20 Preza, G. C., Yang, O. O., Elliott, J., Anton, P. A. & Ochoa, M. T. T lymphocyte density and distribution in human colorectal mucosa, and inefficiency of current cell isolation protocols. *PloS one* **10**, e0122723, doi:10.1371/journal.pone.0122723 (2015).
- 21 Strasner, A. & Karin, M. Immune Infiltration and Prostate Cancer. *Frontiers in oncology* **5**, 128, doi:10.3389/fonc.2015.00128 (2015).
- 22 Cavarretta, I. *et al.* The Microbiome of the Prostate Tumor Microenvironment. *European urology* **72**, 625-631, doi:10.1016/j.eururo.2017.03.029 (2017).
- 23 Sfanos, K. S., Yegnasubramanian, S., Nelson, W. G. & De Marzo, A. M. The inflammatory microenvironment and microbiome in prostate cancer development. *Nat Rev Urol*, doi:10.1038/nrrol.2017.167 (2017).
- 24 Schwartz, D. J., Conover, M. S., Hannan, T. J. & Hultgren, S. J. Uropathogenic *Escherichia coli* superinfection enhances the severity of mouse bladder infection. *PLoS Pathog* **11**, e1004599, doi:10.1371/journal.ppat.1004599 (2015).
- 25 Dardalhon, V. *et al.* IL-4 inhibits TGF-beta-induced Foxp3+ T cells and, together with TGF-beta, generates IL-9+ IL-10+ Foxp3(-) effector T cells. *Nat Immunol* **9**, 1347-1355, doi:10.1038/ni.1677 (2008).
- 26 Chang, H. C. *et al.* PU.1 expression delineates heterogeneity in primary Th2 cells. *Immunity* **22**, 693-703, doi:10.1016/j.immuni.2005.03.016 (2005).
- 27 Pai, S. Y., Truitt, M. L. & Ho, I. C. GATA-3 deficiency abrogates the development and maintenance of T helper type 2 cells. *Proceedings of the National Academy of Sciences of the United States of America* **101**, 1993-1998, doi:10.1073/pnas.0308697100 (2004).
- 28 Zhu, J. *et al.* Conditional deletion of Gata3 shows its essential function in T(H)1-T(H)2 responses. *Nat Immunol* **5**, 1157-1165, doi:10.1038/ni1128 (2004).
- 29 Schmitt, E. *et al.* IL-9 production of naive CD4+ T cells depends on IL-2, is synergistically enhanced by a combination of TGF-beta and IL-4, and is inhibited by IFN-gamma. *Journal of immunology* **153**, 3989-3996 (1994).

- 30 Veldhoen, M. *et al.* Transforming growth factor-beta 'reprograms' the differentiation of T helper 2 cells and promotes an interleukin 9-producing subset. *Nat Immunol* **9**, 1341-1346, doi:10.1038/ni.1659 (2008).

REVIEWERS' COMMENTS:

Reviewer #1 (Remarks to the Author):

My prior comments were all minor and have been addressed adequately in the revised manuscript.

Reviewer #2 (Remarks to the Author):

This is an observation with great potential for prostate cancer therapy. My more pressing concerns have been variously explained away in the response or addressed with changes in wording and additional data. I have many questions about the mechanism, from the perspective of the bacterium, but interesting future work should not delay a decision.

Line 220 perhaps say "Neither monotherapy significantly increased survival of mice, whereas CP1+PD-1 combination therapy conferred a 1.5-fold increased survival ($P = 0.0251$).

Reviewer #3 (Remarks to the Author):

Excellent work and point-by-point reply. No more comments.

March 09, 2018

Dear Reviewers,

Thank you all for your helpful and thoughtful reviews of our manuscript, “*Multi-faceted immunomodulatory and tissue-tropic clinical bacterial isolate potentiates prostate cancer immunotherapy*”. Below, we provide point-by-point responses to the remaining comments:

REVIEWERS' COMMENTS:

Reviewer #1 (Remarks to the Author):

My prior comments were all minor and have been addressed adequately in the revised manuscript.

Thank you for your reviews.

Reviewer #2 (Remarks to the Author):

This is an observation with great potential for prostate cancer therapy. My more pressing concerns have been variously explained away in the response or addressed with changes in wording and additional data. I have many questions about the mechanism, from the perspective of the bacterium, but interesting future work should not delay a decision.

Line 220 perhaps say "Neither monotherapy significantly increased survival of mice, whereas CP1+PD-1 combination therapy conferred a 1.5-fold increased survival ($P = 0.0251$).

Thank you for your reviews, we have updated this sentence in the text (lines 298-300).

Reviewer #3 (Remarks to the Author):

Excellent work and point-by-point reply. No more comments.

Thank you for your reviews.